



# Impact of biomass burning aerosols on radiation, clouds, and precipitation over the Amazon during the dry season: dependence of aerosol-cloud and aerosol-radiation interactions on aerosol loading

Lixia Liu[1], Yafang Cheng[1], Siwen Wang[1], Chao Wei[1], Mira Pöhlker[1], Christopher Pöhlker[1], Paulo Artaxo[2], Manish Shrivastava[3], Meinrat O. Andreae[1,4], Ulrich Pöschl[1] and Hang Su[1]

[1] Max Planck Institute for Chemistry, Mainz, Germany

[2] Institute of Physics, University of São Paulo, São Paulo 05508-900, Brazil

[3] Pacific Northwest National Laboratory, Richland, Washington, USA

[4] Scripps Institution of Oceanography, University of California at San Diego, La Jolla, California, USA

*Correspondence to*: *yafang.cheng@mpic.de & h.su@mpic.de*

**Abstract.** Biomass burning (BB) aerosols can influence regional and global climate through interactions with radiation, clouds, and precipitation. Here, we investigate the impact of BB aerosols on the energy balance and hydrological cycle over the Amazon Basin during the dry season. We performed WRF-Chem model simulations for a range of different BB emission scenarios to explore and characterize nonlinear effects and individual contributions from aerosol-radiation interactions (ARI)

and aerosol-cloud interactions (ACI). The ARI of BB aerosols tend to suppress low-level liquid clouds by local warming and increased evaporation, and to facilitate the formation of high-level ice clouds by enhancing updrafts and condensation at high altitudes. In contrast, the ACI of BB aerosol particles tend to enhance the formation and lifetime of low-level liquid clouds by providing more cloud condensation nuclei (CCN), and to suppress the formation of high-level ice clouds by reducing updrafts and condensable water vapor at high altitudes (> 8 km).

For scenarios representing the lower and upper limits of BB emission estimates for recent years (2002-2016), we obtained total BB aerosol radiative forcings of $-0.2$ W m$^{-2}$ and 1.5 W m$^{-2}$, respectively, showing that the influence of BB aerosols on the regional energy balance can range from modest cooling to strong warming. We find that ACI dominate at low BB emission rates and low aerosol optical depth (AOD), leading to an increased cloud liquid water path (LWP) and negative radiative forcing, whereas ARI dominate at high BB emission rates and high AOD, leading to a reduction of LWP and positive radiative

forcing. In all scenarios, BB aerosols led to a decrease in the frequency of occurrence and rate of precipitation, caused primarily by ACI effects at low aerosol loading and by ARI effects at high aerosol loading.



Overall, our results show that ACI tend to saturate at high aerosol loading, whereas the strength of ARI continues to increase and plays a more important role in highly polluted episodes and regions. This should hold not only for BB aerosols over the Amazon, but also for other light-absorbing aerosols such as fossil fuel combustion aerosols in industrialized and densely populated areas. The importance of ARI at high aerosol loading highlights the need for accurately characterizing aerosol optical properties in the investigation of aerosol effects on clouds, precipitation, and climate.

## 1 Introduction

Biomass burning as a main source of fine particles can influence weather and climate through complex feedbacks with radiation and clouds on regional and global scales (Ramanathan et al., 2001; Kaufman and Koren, 2006; Rosenfeld et al., 2008; Shrivastava et al., 2017; Ditas et al., 2018). Aerosols emitted from biomass burning contain black carbon and brown carbon, which enables them to scatter and absorb solar radiation directly, the so-called 'direct radiative effect' (Charlson et al., 1992; Ackerman et al., 2000). Absorption and scattering of radiation can lead to spatial perturbation and redistribution of energy, therefore trigger subsequent changes in surface energy budget, ground-atmosphere flux exchange, atmospheric thermodynamic stability, and cloud evolution (Li, 1998; Feingold et al., 2005; Cheng et al., 2008a, 2008b; Ding et al., 2013; Huang et al., 2016; Johnson et al., 2004), the so called 'semi-direct effect' (Hansen et al., 1997; Ackerman et al., 2000). These processes, induced by the aerosol radiative effects, are referred to as aerosol-radiation interactions (ARI) (IPCC, 2013). By acting as cloud condensation nuclei (CCN) and ice nuclei (IN) (Crutzen and Andreae, 1990; Roberts et al., 2001; Spracklen et al., 2011), BB aerosols influence the number concentration and size distribution of cloud droplets (Rosenfeld, 2000; Reutter et al., 2009) and thereby change the cloud albedo, i.e., the 'first indirect radiative effect' (Albrecht, 1989; Kaufman and Fraser, 1997), and cloud lifetime, i.e., the 'second indirect radiative effect' (Twomey, 1977; Jiang and Feingold, 2006). The latent heat release that accompanies these internal microphysical processes may modify atmospheric stability and affect convection strength and even subsequent cloud development (Rosenfeld et al., 2008). Such adjustments driven by aerosol microphysical effects are classified as aerosol-cloud interactions (ACI) (IPCC, 2013). Each class of interactions, and their interplay can affect the weather and climate system, leading to enhanced or buffered effects (Tao et al., 2007; Koren et al., 2008; Stevens and Feingold, 2009; Wang et al., 2013).

Mainly driven by deforestation and agricultural practices (Echalar et al., 1998; Reddington et al., 2015), biomass burning events prevail in the Amazon Basin (Setzer and Pereira, 1991) during the dry season, typically between July and October (Gan et al., 2004), injecting large amounts of aerosols into the atmosphere. Particle numbers during the peak of the burning season in the Amazon may increase one order of magnitude compared with the concentration levels during seasons without biomass fires (Martins et al., 1998; Andreae et al., 2002; Roberts et al., 2003; Martin et al., 2010). As most of the Amazon region is located in the equatorial and subequatorial area with the Intertropical Convergence Zone passing across it, the radiation budget



and convection system there play important roles in the global energy balance, carbon storage, and transport of water vapor (Sengupta et al., 1990; Bony et al., 2006; Su et al., 2011) and aerosols (Freitas et al., 2005). During the dry season, precipitation

amounts are relatively low, rendering the rainforest ecosystem more vulnerable to rainfall changes. Therefore, perturbations imposed by BB aerosols during the dry season are important for climate and ecology in Amazonia and even globally (Andreae et al., 2004).

Extensive investigations regarding the influence of BB aerosols on radiation and convection in this region by observation (Williams et al., 2002; Andreae et al., 2004; Lin et al., 2006; Goncalves et al., 2015; Braga et al., 2017; Cecchini et al., 2017)

and modelling studies (Feingold et al., 2005; Liu, 2005; Zhang et al., 2008; Wu et al., 2011b; Ten Hoeve et al., 2012; Kolusu et al., 2015) have been conducted. BB aerosols were reported to cause a negative direct radiative forcing ranging from several to tens of W m$^{-2}$ at the top of atmosphere (TOA) over the Amazon area (Procopio et al., 2004; Zhang et al., 2008; Sena et al., 2013; Kolusu et al., 2015). Yet, their total radiative forcing varies in sign and magnitude between different modelling estimates (Ten Hoeve et al., 2012; Kolusu et al., 2015; Archer-Nicholls et al., 2016) because of uncertainties associated with the

prescription of aerosol optical properties, cloud sensitivity to BB aerosols, model resolution (Archer-Nicholls et al., 2016), etc. BB aerosols over the Amazon were observed to efficiently increase cloud droplet number and decrease cloud droplet radius (Andreae et al., 2004; Cecchini et al., 2017). However, satellite remote sensing measurements showed both suppression and enhancement of cloud fraction with the presence of BB aerosols in the Amazon (Kaufman and Fraser, 1997; Koren et al., 2004; Kaufman and Koren, 2006; Koren et al., 2008), and suggested a dependence of cloud response on aerosol concentrations

(Koren et al., 2008). Simulations by both cloud-resolving models and regional atmosphere-aerosol coupled models found enhanced cloud water burdens due to the microphysical effects of BB aerosols (Wu et al., 2011b; Reutter et al., 2014; Chang et al., 2015). Their radiative effect was shown by large-eddy simulation to efficiently diminish liquid cloud amount by evaporating cloud droplets and suppressing vapor availability from land-atmosphere flux exchange (Feingold et al., 2005). Precipitation from convective clouds was also reported to be either inhibited or invigorated based on observations from in-situ,

aircraft, and satellite remote sensing measurements (Andreae et al., 2004; Lin et al., 2006; Goncalves et al., 2015). Cloud-resolving modelling found nonlinear relationships between aerosol loading and precipitation through the microphysical effects of BB aerosols (Carslaw et al., 2013; Chang et al., 2015). Regional modelling studies showed that their radiative effect could cause an overall reduction in precipitation, but may increase the nighttime precipitation (Wu et al., 2011b) and intensify the extreme high precipitation rates (Kolusu et al., 2015).

Inter-annual variability is a prominent characteristic of the biomass burning intensity in the Amazon (Kaufman and Fraser, 1997; Bevan et al., 2009; Pöhlker et al., 2019). However, most previous studies assessed the climate response to the perturbation from BB aerosols based on the emission scenario of one specific year (Zhang et al., 2008, 2009; Wu et al., 2011b; Ten Hoeve et al., 2012; Kolusu et al., 2015; Archer-Nicholls et al., 2016). Given the possible nonlinear relationship between convection and aerosol concentration and the sensitivity of aerosol radiative forcing to aerosol loading, the necessity of a

thorough assessment of radiation, clouds, and precipitation response to BB aerosols over an extensive range of emissions is



underscored. Although Thornhill et al. (2018) estimated the difference in cloud response between high and low emission intensity scenarios, this difference may not be adequate to serve as constraint for estimating BB aerosols' impact on background Amazon climate, since the perturbations due to BB aerosols may be nonlinear and have been proven to be strongly dependent on the reference emission setting (Wang, 2005; Martins et al., 2009). In this study, we performed WRF-Chem simulations over

the Amazon Basin in September 2014 with a 'clean' condition, defined by the absence of influence from biomass burning, and a set of emission scenarios resembling the realistic inter-annual emission variability in the dry season, to investigate the effects of BB aerosols on the Amazon radiation budget, clouds and precipitation quantitatively and mechanistically. Comparison of the precipitation in central Amazonia in the year 2014 with that averaged over 18 years (1998–2016) indicates that the atmospheric conditions in this region in 2014 are climatically representative (Pöhlker et al., 2016). Therefore, the present study

based on September 2014 may serve to represent the typical sensitivity behavior of the dry season climate to BB aerosol concentration variations. Climatically significant estimates of BB aerosols' radiative forcing, which may require statistics of over 30 years (Fiedler et al., 2017), are out of the scope of this study. As case study simulations imply that the initial convection response may influence secondary convection (Khain et al., 2005), monthly averaged effects of BB aerosols were assessed here to demonstrate an overall characteristic for the whole month. Individual processes of ARI and ACI were disentangled in

our simulations, based on which the relative significance of the two pathways and their sensitivity to emission intensity were quantified. In this paper, the model description and experiment design are documented in Sect. 2. An evaluation of the simulation results for the meteorological and chemical fields against observations is presented in Sect. 3. The overall impacts of BB aerosol emissions on radiation, meteorological conditions, clouds, and precipitation are shown in Sect. 4. Conclusions are in Sect. 5.


## 2 Model and data

### 2.1 Model description

WRF-Chem is an online-coupled meteorology-chemistry model, which integrates meteorology and chemistry with aerosol-radiation-cloud feedbacks (Grell et al., 2005). WRF-Chem version 3.9.1 was used in this study to investigate the impact

of BB aerosols on the energy budget and hydrological cycle over the Amazon Basin.

The Carbon-Bond Mechanism version Z (CBMZ) gas-phase chemistry mechanism (Zaveri and Peters, 1999) and the Model for Simulating Aerosol Interactions and Chemistry (MOSAIC) aerosol module (Zaveri et al., 2008) were selected. The aerosol size distribution is described by 8 discrete size bins defined by their lower and upper dry particle diameters ranging from 39 nm to 10 μm. Aerosols are assumed internally mixed in each bin to engage in microphysical processes. To participate

in the radiative processes, each aerosol component is prescribed with a refractive index based on the value suggested in Barnard (2010). To avoid the overestimation of the particle absorption cross-section when using the internal mixing of BC with other aerosol components (Bond and Bergstrom, 2006), the Maxwell-Garnett mixing rule assuming spheres of BC distributed



randomly throughout a mixture of other aerosol components was applied in this study (Bond and Bergstrom, 2006). With the mixed refractive indices, the aerosol extinction efficiency, single-scattering albedo, and asymmetry factor are computed using a Mie algorithm for each size bin and wavelength. The total optical properties are then obtained by integrating over all of the size bins and used as inputs to the RRTMG radiation transfer model for the shortwave (Fast et al., 2006) and longwave spectrum (Zhao et al., 2013). Aerosol-cloud interactions are accounted for in the model through three pathways: activation of aerosol particles to form cloud droplets as well as their resuspension from evaporating cloud droplets, aqueous chemistry, and wet deposition (Chapman et al., 2009). Aerosols are treated as 'interstitial' or 'cloud-borne' according to whether they are activated as CCN, and the calculation of the activation process follows the methodology of Abdul-Razzak (Abdul-Razzak and Ghan, 2002). The two-moment Lin microphysics scheme (Lin et al., 1983; Rutledge, 1984) was employed in this study, where prognostic cloud droplet number is treated based on activated aerosols following Ghan et al. (1997) and the autoconversion of cloud droplets to rain droplets is dependent on droplet number (Liu et al., 2005) so that aerosols are allowed to potentially influence the rain rate and liquid clouds (Ghan et al., 1997; Chapman et al., 2009). The aerosol-aware Lin microphysics scheme has been used previously in investigating aerosol impacts on synoptic cyclones (Ye et al., 2019), regional fog (Lee et al., 2016b), and local convection systems (Wu et al., 2011b). In order to validate the response of our model to increasing CCN, monthly mean domain-averaged cloud droplet radii and corresponding cloud-base CCN concentrations were calculated for simulations with different emission rates, shown in Fig. S5. The sensitivity of cloud droplet radius to increasing CCN concentration is pronounced at lower CCN concentrations, while the response tends to saturate at higher CCN concentrations. The saturation of the response of droplet radius to aerosol concentration has also been observed by satellite (Breon et al., 2002). These observations suggested a saturation point at aerosol optical depth (AOD) of 0.3, which corresponds to the relatively higher aerosol concentration scenario (EMIS3) in our study. 'Cloud-borne' aerosols and trace gases dissolved in cloud water interact through aqueous chemistry, which may modify aerosol composition and content. The aqueous-phase chemistry is based on the Carnegie-Mellon University (CMU) bulk aqueous phase chemical mechanism (Fahey and Pandis, 2001). Wet deposition of aerosols includes in- and below cloud removal through being collected by rain, graupel, and snow (Chapman et al., 2009) and through being scavenged by precipitation washout (Easter et al., 2004), respectively. Other major schemes utilized, e.g., the RRTMG longwave and shortwave radiation scheme (Mlawer et al., 1997; Pincus et al., 2003), the Yonsei University (YSU) boundary layer scheme (Hong, 2010), the Rapid Update Cycle (RUC) land surface scheme (Smirnova et al., 1997; Smirnova et al., 2000), the Grell-Devenyi cumulus parameterization (Grell and Devenyi, 2002), and the Fast-J photolysis rate scheme (Wild et al., 2000), are described in Table 1.

In this study, three nested domains with horizontal resolutions of 75 km, 15 km, and 3 km were set up over South America, as shown in Fig. 1. Domain1 covers most of the South American continent, with the biomass burning source region included. Domain3 centers around the ATTO site to represent the typical climate and environment of the central Amazon Basin (Andreae et al., 2015), and uses cloud-resolving grid spacing with Grell cumulus parameterization turned off (Table 2). Vertical layers of 29 levels extending from ground to 50 hPa were employed for all domains. The outer domains were two-way coupled with initial and boundary meteorological and chemical conditions from the 6-hour National Centers for Environmental Prediction





(NCEP) Final Analysis (FNL) data and Model for Ozone and Related Chemical Tracers, version 4 (MOZART-4) global chemical transport model output (Emmons et al., 2010), respectively. The Four Dimensional Data Assimilation (FDDA) of temperature, horizontal wind and moisture was applied for the outer domains to reduce meteorological biases (Otte et al.,

2012). The innermost domain was driven one-way by initial and boundary inputs from the outer domain. Anthropogenic emissions were from the EDGAR-HTAPv2, a global, gridded air pollution emission dataset with a resolution of 0.1° × 0.1°. The biogenic emissions were generated online by Model of Emissions of Gases and Aerosols from Nature (MEGAN) (Guenther et al., 2006). The Fire Inventory from NCAR version 1.5 (FINNv1.5) (Wiedinmyer et al., 2011), which provides global estimates of the trace gas and particle emissions from open fires updated daily with 1 km resolution, was used to provide

the biomass burning emissions. The primary organic matter (POM) emission rate was converted from OC emission based on an observed ratio of 1.5 between the mass of POM and OC (Reid et al., 2005). The conversion factor 1.5, broadly used in WRF-Chem simulations for biomass burning emission (Ge et al., 2014; Archer-Nicholls et al., 2015), represents the lower end of the range of POM/ OC ratios for fresh aerosol emissions from biomass burning (Yokelson et al., 2009; Takahama et al., 2011; Brito et al., 2014; Collier et al., 2016; Andreae, 2019). Plume ascent from fire emission is calculated by a plume rise

parameterization (Grell et al., 2011; Freitas et al., 2007). The simulation was conducted from 24 Aug to 30 Sep 2014, when the Amazon Basin was undergoing its dry season with biomass burning prevalent. The first 6 days of the simulation were used as spin up. Details on model configurations are listed in Table 1.

## 2.2 Design of numerical experiments

In order to quantitatively investigate the impact of BB aerosols on radiation, cloud, and precipitation, a set of BB aerosol emission scenarios generated by multiplying different aerosol emission factors (X) with the original BB aerosol emission scenario was applied to all domains. As sub-grid convective parameterization can cause uncertainties to the impacts from BB aerosols due to the lack of aerosol-cloud interactions in the sub-grid convective parameterization (Archer-Nicholls et al., 2016), the analysis of BB aerosols' effects in Sect. 4 is based on the domain3 simulation where convections are explicitly resolved at

3 km resolution. Simulations of domain3, namely PC3_EMISX, were conducted using the BB aerosol emission scenario (EMISX) and chemical boundary conditions from the outer-domain simulation with the corresponding emission scenario. A control simulation CC3 was conducted without influence of biomass burning emissions. Then the BB aerosol total effect can be evaluated from the difference between the PC3_EMISX and CC3 simulations.

   To distinguish aerosol-cloud interactions (ACI) effect from aerosol-radiation interactions (ARI) effect, we adopted a

similar method as in Wu et al. (2011). Parallel simulations with PC3_EMISX and CC3 were performed without aerosol radiative feedbacks, namely PCNR3_EMISX and CCNR3 (Table 2). The aerosol-cloud interactions (ACI) or aerosol microphysical effect of each BB aerosol emission scenario can be assessed from the difference between PCNR3_EMISX and CCNR3. Then the aerosol-radiation interactions (ARI) effect could be obtained by deducting the ACI effect from the aerosol total effect (Archer-Nicholls et al., 2016).





As shown in Fig. 2, the biomass burning emissions during September undergo remarkable annual variations, e.g., the emission in 2007 is 6 times as much as that in 2014. The variation pattern of $PM_{10}$ emitted from BB in September is consistent with an inter-annual variation of MODIS retrieved AOD over the Amazon (Sena et al., 2013). Based on the range of emission intensities from 2002 to 2016, we set three emission scenarios representing different emission strength: EMIS1 for emission in 2014, EMIS3 for an average intensity over all years, and EMIS6 for the emission intensity in 2007, which corresponds to

the maximum emission intensity from 2002 to 2016. An addition, emission scenario EMIS0.5 was added to mimic the reduced BB emissions projected assuming the influence of enhanced government regulation policy (Streets, 2007). The domain3-averaged AOD in the simulations for the EMIS0.5, EMIS1, EMIS3, and EMIS6 emission scenarios is used to represent the aerosol concentration under the corresponding emission scenarios in the analysis in Sect. 4.

### 2.3 Data

**2.3.1 Ground based measurements at ATTO**

    The Amazon Tall Tower Observatory (ATTO) site is located in the central Amazon, about 150 km northeast of Manaus, Brazil (Fig. 1), and represents a relatively clean rainforest background environment (Andreae et al., 2015). An 80-m tower, embedded within the canopy of about 35 m height, provides continuous measurements related to the research fields of meteorology, atmospheric trace gases, and aerosols (Andreae et al., 2015). Observational datasets used in this paper include

meteorological variables, cloud condensation nuclei (CCN) number concentrations, and black carbon mass. Meteorology observations were obtained from a thermohygrometer and a 2-D sonic anemometer installed at 55 m on the tower. The data was averaged at 10-min resolution from raw observations taken at 1-min resolution. Air temperature and relative humidity (RH) measurements are only available from 1 Sep to 23 Sep 2014, while wind speed was observed over the complete simulation period. Aerosols were sampled at 60 m height. The CCN number concentration measurements by a CCN counter with

supersaturation cycling through a set of levels ranging from 0.11% to 1.1% were used in this study. Detailed information on the CCN dataset can be found in Pöhlker et al. (Pöhlker et al., 2016, 2018). Equivalent black carbon (BCe) mass concentrations, $M_{BCe}$, were obtained from a multi-angle absorption photometer (MAAP) ($\mu g\ m^{-3}$). The mass concentration was calculated by dividing the absorption coefficients at 637 nm by the dry season mass absorption cross-section of 12.3 $m^2\ g^{-1}$ according to Saturno et al. (2018a). Specific details about the MAAP measurements of $M_{BCe}$ can be found in Saturno et al. (2018a).


    **2.3.2 Radiosonde measurements**

    Radiosonde observations from the Integrated Global Radiosonde Archive (IGRA) (Durre et al., 2006) at the site Manaus (3.15 °S, 59.98 °W) were used to evaluate the vertical profile of meteorological elements. The IGRA sounding observations were conducted at 00:00 and 12:00 Universal Standard Time (UTC, which is 4 hours before local time). Note that throughout

this paper the time referred to is UTC, unless Local Time (LT) is specifically mentioned. Radiosonde measurements at standard





pressure levels within the troposphere (100, 150, 200, 250, 300, 400, 500, 700, 850 and 1000 hPa), as well as derived characteristic indices for convection, e.g., Convective Available Potential Energy (CAPE) and Lifting Condensation Level (LCL), were compared with simulation outputs.

### 2.3.3 TRMM

The Tropical Rainfall Measuring Mission (TRMM) 3B42 (Huffman et al., 2007) product was employed to evaluate the precipitation simulation performance. TRMM provides satellite observations of tropical and subtropical (50 °S–50 °N) precipitation globally. The 3B42 rainfall dataset produces 3-hour averaged high-quality, infrared and microwave precipitation estimates at a resolution of 0.25° × 0.25° (Huffman et al., 2007).

### 2.3.4 MODIS

The Moderate Resolution Imaging Spectroradiometer (MODIS) Level 3 products provide satellite-derived daily estimates of AOD and cloud properties at a resolution of 1° × 1° (Remer et al., 2005). The measurements by MODIS onboard Aqua were used in this study, as the passing time of Aqua is approximately at 13:30 LT, when the local convective system is highly developed (Koren et al., 2004). The retrieval of AOD at 550 nm was used to validate the simulated aerosol concentration, and observations of cloud fraction, total liquid water path (LWP), and total ice water path (IWP) were compared with the corresponding modelling results. Model outputs at the satellite detection time were used when comparing against MODIS data.

### 2.3.5 AERONET

The Aerosol Robotic Network (AERONET) is an observation network of about 100 sites distributed globally, providing continuous measurements of aerosol optical properties (Holben et al., 2001). The observations are made by sun- and sky-scanning ground-based automated radiometers at various wavelengths. AOD and single scattering albedo (SSA) at 550 nm were interpolated using corresponding measurements at 675 nm and 440 nm. The level 2.0 cloud-screened product is used in this study. Data is retrieved for three sites during September 2014 within the Amazon Basin: Ji_Paraná_SE (10.9 °S, 61.9 °W), Manaus_EMBRAPA (2.9 °S, 60.0 °W) and Alta_Floresta (9.9 °S, 56.1 °W).





## 3 Model evaluation

To validate the model's capability to represent and quantify BB aerosol effects on regional climate through aerosol-radiation-cloud feedbacks, the WRF-Chem simulation with the EMIS1 scenario is evaluated by comparing with ground-based
and satellite remote sensing measurements of the meteorological and aerosol fields.

### 3.1 Meteorological condition

Figure 3 shows time series of hourly surface meteorological variables observed at the ATTO site and corresponding simulated results from domain3 in September 2014. As the canopy effect is integrated in the land surface model (Lee et al.,
2016a), the simulated meteorological variables are characterized by above-canopy properties. The variation patterns of the air temperature and RH are well captured by the model, with correlation coefficients of 0.86 and 0.78, respectively (Table 3). The surface air temperature is reproduced with a moderate overestimation of 0.2 °C (Table 3), which mainly occurs on 6 Sep, 8 Sep, and 17–18 Sep, whereas the RH exhibits an opposite bias. These significant biases occur corresponding to the missing prediction of rainfall on 6 and 18 Sep and underprediction of precipitation on 8 and 17 Sep by the model at this site (Fig. S1).
Hence, a bias of rainfall location and existence of unresolved subgrid-scale (<3 km) convective precipitation may account for the discrepancy between simulated and observed surface temperature and RH. The wind speed from the simulation is generally lower than the observations with an average bias of –0.2 m s⁻¹ (Table 3). The underestimation of the surface wind speed by the model also existed extensively in previous WRF-Chem simulations, and was ascribed to uncertainties in surface drag parameterization (Tuccella et al., 2012; Zhang et al., 2015).
The vertical distribution of the meteorological variables at the Manaus site over the 30-day simulation period is compared in Fig. 4. To keep consistency, simulation outputs of temperature and RH from domain3 were interpolated to the standard levels of the radiosonde data. The convective available potential energy (CAPE) and lifted condensation level (LCL), inferred from the temperature and humidity profiles from modeling and observations, are also shown. The model reproduces the air temperature profiles well. The RH generally follows the observed results below 300 hPa, while in the upper troposphere above
300 hPa a large overestimation occurs at 12:00 UTC. Similarly, an overestimation of simulated water vapor compared with MLS retrievals in the upper troposphere was found in WRF-Chem simulations of the Amazon Basin (Wu et al., 2011a). The CAPE and LCL values estimated from the model agree well with that from the observations at 00:00 UTC. Noticeable differences of CAPE and LCL between model and observation of 240 J kg⁻¹ and 266 m, respectively, are seen at 12:00 UTC, implying a possible earlier development of the simulated planetary boundary layer (PBL) ahead of observations.
The daily retrievals of cloud fraction, total LWP, and total IWP from the MODIS Aqua measurements are used to evaluate the simulation performance for cloud properties by WRF-Chem. The domain3 simulation results are averaged over the domain area to compare with the corresponding variables from the satellite measurements, as shown in Fig. 5. The simulated total LWP, calculated as the sum of liquid cloud and rainwater, correlates well with observations with a moderate underestimation.


The total IWP from the model, as the sum of cloud ice, snow, and graupel, basically shows a positive correlation with the

observations. However, a large underestimation of the total IWP from the model exists compared to the remote-sensed data. The model performs relatively well for the extreme low and high IWP regimes, with values being approximately 25% of the observations. The simulation of the total IWP by the WRF model has been found to produce a seasonally averaged underestimation by up to 80% compared with satellite measurements (Baro et al., 2018), suggesting uncertainties in the ice-phase microphysical processes. The lack of IN parameterization in the microphysical scheme may contribute to this

discrepancy (Su et al., 2018). Generally, the total cloud fraction from the model shows a linear correlation with the observations, falling between 25%–75% of the observed values.

Figure 6 shows the time series of domain-averaged 3-hour accumulated precipitation from the domain3 simulation and corresponding TRMM measurements during September 2014. The model well captures the occurrence of rainfall measured remotely by satellite. The regional rainfall events on 6, 8, 17 and 18 Sep are well predicted by the model with a slight

underestimation, corroborating the hypothesis that the bias of surface air temperature and RH are likely induced by a local bias of rainfall location and neglecting precipitation of sub-grid convection by the model. Generally, the simulated precipitation is comparable with TRMM observations in terms of time variation and intensity, which illustrates the model's ability to represent the convective activity during the study period.

**3.2 Aerosols**

The simulated AOD is evaluated by comparison with observations from AERONET and MODIS at three AERONET stations (Fig. 7). Simulation results are extracted from the domain2 simulations and the MODIS data is from the corresponding pixel from daily retrieval datasets. Since the Alta_Floresta and Ji_Paraná_SE sites are located closer to the southern border of the Amazon Basin, where biomass burning takes place, they show higher AOD than Manaus_EMBRAPA, with ground-based

instantaneous observation maxima of 0.9, 0.8, and 0.4 for Alta_Floresta, Ji_Paraná_SE, and Manaus_EMBRAPA, respectively. The simulated AOD is comparable with the observations at Manaus_EMBRAPA, with peaks occurring on 25 and 26 Sep that could have been influenced by BB plumes. The AOD at Alta_Floresta and Ji_Paraná_SE is underestimated, especially for high-value episodes. The error in detection of fire by satellite may result in an underestimated BB emission and therefore an underpredicted AOD (Rosario et al., 2013), which is seen by the difference between AERONET and MODIS on

14 Sep at Alta_Floresta. Besides, the lack of SOA production in the model may lead to an underestimation not only due to its direct contribution to light extinction but also because the presence of OC coatings may amplify the light absorption by BC (Bond and Bergstrom, 2006). The negative bias in simulating RH may also cause a lower AOD owing to the dependence of AOD on RH (Robert et al., 1998).

The simulated SSA is compared with AERONET retrievals for Alta_Floresta and Ji_Paraná_SE and observations from

previous studies in Table 4. The comparison at Manaus_EMBRAPA is excluded because of the scarcity of the data available. The simulated SSA is lower compared with AERONET retrievals for Alta_Floresta and Ji_Paraná_SE. However, the SSA





observed on flights at Porto Velho, a site at the southwestern Amazon, during the SCAR-B campaign (Reid et al., 1998), is close to the simulated result. Compared with the in-situ measured SSA of 0.87±0.06 at 637 nm at the TT34 tower (Rizzo et al., 2013) in the central Amazon, a higher value of 0.90±0.01 is obtained by our model simulation. Similarly, the modeled

monthly mean SSA of 0.90 for the location of the ATTO site is slightly higher than an extrapolated value of 0.88 at 550nm from multi-year observations for the dry season at the ATTO site (Saturno et al., 2018b). Given the substantial influence of BB particles on the aerosol SSA (Saturno et al., 2018b), the difference between model results and observation may also possibly be caused by the mismatched average time periods for the comparison. In Fig. S2 we compare the time variation of black carbon mass concentration measured at the ATTO site during the simulation period with the simulation outputs. The

model results are in fair agreement with the observed BC concentrations, indicating a reasonable estimate of the influence from BB on this region.

A comparison of CCN concentrations at different supersaturations between in-situ observation and the WRF-Chem simulation for the ATTO site is presented in Fig. 8. The calculation of CCN number concentration at observed supersaturation level from model outputs followed the method in Su et al. (2010). The model results show an overall agreement in magnitude

with observations for the supersaturation range of 0.2-0.5%, which represents the typical atmospheric conditions during the dry season in the Amazon (Archer-Nicholls et al., 2016). The variation of CCN number with supersaturation level matches the pattern obtained by observation (Pöhlker et al., 2018), indicating a reasonable sensitivity of aerosol activation ability to varying supersaturation situations.

**4 Results**

**4.1 Impact on radiation**

Figure 9 shows the diurnal cycle of the BB aerosol impact on the domain-averaged all-sky shortwave radiation based on the EMIS1 emission scenario. The ACI effect, in which BB aerosols act as CCN, causes negative radiative perturbations to shortwave radiation both at TOA (Fig. 9a) and the surface (Fig. 9b) during the daytime. This can be attributed to the increased

cloud albedo as a result of larger cloud LWP and smaller cloud droplet radius (Table 5) caused by the ACI (Twomey et al., 1977). The magnitude of $-0.7$ W m$^{-2}$ for shortwave radiative forcing (RF) at TOA is comparable to the global indirect aerosol forcing of $-1.7$ to $-1.0$ W m$^{-2}$ by carbonaceous aerosols from fires estimated by Ward et al. (Ward et al., 2012).

The radiation perturbations due to the ARI effect are more complicated as they involve the direct radiative effect of BB aerosols themselves and subsequent cloud adjustments. Fig. 9 and S3 show a clear difference in radiative forcing with and

without considering clouds (all-sky versus clear-sky conditions). In clear-sky cases, BB aerosols reduce the shortwave radiation reaching the ground by directly scattering and absorbing incident solar radiation, leading to a reduction of shortwave radiation at the surface of $-6.7$ W m$^{-2}$ (Fig. S3b). The clear-sky shortwave RF by ARI at TOA is negative for most of the day except at local noon (15:00 UTC to 17:00 UTC) when the PBL fully develops (Fig. S3a). Such a diurnal variation can be





explained by the evolution of aerosol vertical distributions. The vertical location of absorbing aerosols is an important
controlling factor for their absorptivity (Samset and Myhre, 2011). When aerosols are lifted higher by the vigorously grown
PBL, the absorption of solar radiation by BB aerosols is amplified resulting in more heating and positive forcing. On average,
the clear-sky shortwave RF by ARI at TOA is about –0.7 W m$^{-2}$, and corresponds to a cooling effect on the Earth-atmosphere
systems, which is consistent in sign with observational and modelling results in this region (Sena et al., 2013; Archer-Nicholls
et al., 2016; Thornhill et al., 2018). When taking clouds into consideration, the all-sky shortwave radiative perturbation by
ARI is about –5.7 W m$^{-2}$ and 0.4 W m$^{-2}$ at the surface and TOA (Table 5), respectively. Compared with the clear-sky results,
the positive shifts of radiative perturbation by ARI in all-sky condition at both the surface and TOA indicate less solar radiation
reflected back to space. This can be accounted for by the decreased liquid cloud water content (Table 5) due to the BB aerosols'
radiative effect, which results in more incident solar radiation (so-called 'semi-direct' effect). Seen from the diurnal cycle of
shortwave forcing by the ARI (Fig. 9a), the time period when the radiative forcing is positive becomes longer, although
negative values still exist in the early morning and late afternoon when the cloud response is negligible (Fig. 13b). In previous
studies, the positive radiative forcing associated with the reduction of cloud cover was shown to be very strong (Zhang et al.,
2008), even to the point of being able to reverse the sign of the BB aerosols' direct radiative forcing over the Amazon (Koren
et al., 2004; Archer-Nicholls et al., 2016).

The total shortwave RF at TOA by BB aerosols is a result of the competing ACI and ARI effects. Fig. 10a shows the total
shortwave RF caused by BB aerosols from emission scenarios with different aerosol emission intensities (represented as the
domain-averaged AOD). The relative importance of the ACI and ARI effect on shortwave RF varies with the aerosol loading
under the same atmospheric conditions. The total shortwave forcing is negative at lower aerosol loading, dominated by the
ACI effect, but shifts to be positive at higher aerosol loading, driven by the ARI effect. This is expected because changes in
the aerosol loading will change the cloud properties more at low background aerosol concentrations than at higher aerosol
abundance where the effects tend to be saturated (Twomey, 1977; Roberts et al., 2003); and the ARI effect associated with
aerosol extinction of radiation intensifies with increasing aerosol concentration (Koren et al., 2004).

Such nonlinear ACI and ARI effects of BB aerosols are consistent with their effects on the cloud water (Fig. 12a),
implying the importance of cloud adjustments for affecting BB aerosol RF. At TOA, the monthly mean shortwave RF by BB
aerosols (ACI + ARI) is –0.3 W m$^{-2}$ and 0.6 W m$^{-2}$ for the EMIS1 and EMIS6 scenarios (Table 5), respectively. Similar in
magnitude, a modelling study of the Amazon dry season using the HadGEM3-GA3 model showed a monthly mean shortwave
RF of 1.35 W m$^{-2}$ with AOD increasing from 0.19 to 0.67 (Thornhill et al., 2018). The longwave RF by BB aerosols is of
comparable magnitude to the shortwave radiative forcing (Table S1). The ARI is the driving force for the positive longwave
RF, as the outgoing infrared radiation can be directly trapped by black carbon contained within the BB particles (Ramachandran
and Kedia, 2010). Besides, high clouds mainly comprised of ice are also efficient in blocking outgoing longwave radiation
(Hartmann et al., 1992), yielding a positive longwave RF at TOA. Therefore, the ARI-induced larger amount of cloud ice
content (Fig. 13b) can result in positive longwave RF as well. The positive longwave RF resulting from increased ice cloud is
in agreement with the satellite observations of tropical deep convection, where a strong warming was caused by increased





convective cloud anvils impacted by aerosols (Koren et al., 2010). However, it should be noted that, as the ice cloud response
is a crucial factor for determining the longwave RF, the lack of parameterization of the aerosols' role as IN adds uncertainty
to the simulated longwave RF by BB aerosols. The appreciable magnitude of longwave RF (Thornhill et al., 2018; Archer-
Nicholls et al., 2016) underlines the necessity of further studies to constrain the BB aerosol effect on high clouds. The all-band
RF (shortwave plus longwave) of BB aerosols changes sign with increasing emission intensity of BB aerosols, with values of
$-0.2$ W m$^{-2}$ and 1.5 W m$^{-2}$ for the EMIS1 and EMIS6 scenarios, respectively.

        At the surface, a reduction in shortwave radiation is induced by the presence of BB aerosols, which intensifies with higher
emission intensity. Compared with previous model estimates, a $-15.9$ W m$^{-2}$ shortwave reduction estimated from a multi-day
biomass burning simulation in 2006 using WRF-Chem (Wu et al., 2011b) is of similar magnitude as the $-17.1$ W m$^{-2}$ in this
study using the EMIS3 scenario, which is almost equivalent to the emission intensity of the year 2006. However, the magnitude
of the estimates diverges in different models, e.g., $-28.2$ W m$^{-2}$ was induced by an increase of AOD by about 0.4 using the
GATOR-GCMOM model (Ten Hoeve et al., 2012), and $-5.46\pm1.93$ W m$^{-2}$ was calculated with an increase of AOD by about
0.5 using HadGEM3-GA3 (Thornhill et al., 2018), which may result from different parameterizations of the aerosol optical
properties and treatments of cloud response. The decreased solar radiation at the surface is balanced mostly (over 90%) by a
reduced sensible and latent heat flux (Table 5) and marginally by the earth-emitted infrared radiation. Specifically, the ARI
leads to a decrease of $-2.9\%$ ($-17.6\%$) and $-2.0\%$ ($-12.0\%$) for sensible heat and latent heat, respectively, in the EMIS1
(EMIS6) scenarios, which could impose an inhibiting effect on cloud formation (Yu et al., 2002; Feingold et al., 2005; Jiang
and Feingold, 2006; Rosenfeld et al., 2008).

### 4.2 Impact on atmospheric stability

        Figure 11 shows the diurnal and vertical distribution of the BB aerosol impact on the domain-averaged air temperature,
RH, and vertical velocity for the EMIS1 scenario. Prominent responses of air temperature and RH occur below 5 km, where
aerosols concentrate. Affected by the ARI, air temperature is reduced within the PBL, but is increased at the top of the PBL
(Fig. 11b). As aforementioned, BB aerosols reduce the incident solar radiation at the surface and therefore decrease the heat
flux from the ground to the atmosphere. Consequently, the air temperature within the PBL drops, with a diurnal maximum
reduction of over $-0.05$ °C near the surface. The solar radiation absorbed by the black carbon in BB aerosols heats the
atmosphere (Bond et al., 2006), producing an increase in air temperature by about 0.03 °C near the top of the PBL. This vertical
distribution of temperature responses tends to stabilize the PBL and suppresses the upward velocity (Fig. 11b). On the other
hand, the increased air temperature at the top of the PBL destabilizes the air above and stimulates updrafts (Feingold et al.,
2005; Koch and Del Genio, 2010). The intensified upward airflow delivers more water vapor to higher altitudes, leading to a
pronounced increase in the RH at altitudes above 10 km. The ACI effect acts opposite to the ARI effect in changing the
thermodynamic structure. The air cools at the top of the PBL, since more evaporation-induced cooling is generated with more
but smaller cloud droplets (Table 5). In contrast, higher air temperatures within the PBL can be the result of less evaporative





cooling from precipitating hydrometeors (Fig. 14a). Overall, the thermodynamic response to BB aerosols is dominated by the ARI effect. The diurnal mean change of surface air temperature is –0.2 °C in the EMIS6 scenario (Table 5), in agreement with other modeling results for the Amazon area (Kolusu et al., 2015; Thornhill et al., 2018).

**4.3 Impact on cloud**

Figure 13 shows the diurnal and vertical distribution of domain-averaged changes in cloud water and cloud ice concentration caused by BB aerosols. By serving as CCN, BB aerosols create more cloud droplets and cause a reduction in the cloud droplet size (Table 5) due to competition for water vapor, which slows down the transfer rate from cloud to rain (Rosenfeld et al., 2008; Chang et al., 2015; Braga et al., 2017). Consequently, cloud water in the free troposphere is increased by the ACI effect throughout the day (Fig. 13a) at the expense of rainwater concentration, while the diminished cloud water within the PBL corresponds to the warmer air temperature (Fig. 11a) and suppressed moisture flux from the ground surface (Table 5).

The response of cloud water to the ARI also varies with altitude. The increased RH within the PBL by the radiative effect (Fig. 11b) lowers the cloud base height (Table 5) and favors cloud persistence, resulting in higher cloud water content (Johnson et al., 2004). In contrast, the radiative heating near the top of the PBL (Fig. 11b) decreases the RH and therefore 'burns off' the liquid clouds (Feingold et al., 2005; Huang et al., 2016). Such contrasting cloud water response to the BB aerosol radiative effect between different layers was also found by a large-scale RegCM3 simulation covering South America (Zhang et al., 2008). The increased cloud water in the lower troposphere (0–2 km) was attributed to large-scale moisture convergence. Here, the simulation over a smaller region located in the central Amazon Basin shows that the local modification of the thermodynamic structure by BB aerosols is able to contribute to the effect as well.

Integrated over the atmosphere, the cloud LWP is enhanced by the ACI, but reduced by the ARI effect (Ackerman et al., 2000; Johnson et al., 2004; Feingold et al., 2005). Therefore, the overall change of cloud water amount by BB aerosols results from the competition between the ACI and ARI effects. Figure 12 displays the dependence of the overall response of cloud water on the emission intensity (represented as AOD). Weaker emission scenarios yield higher cloud water, driven by the ACI effect, while stronger emissions lead to an opposite response of cloud water, dominated by the ARI effect. The simulated dependence of cloud water change on aerosol amount agrees with satellite measurements of the total cloud fraction over the Amazon region (Koren et al., 2004).

The cloud ice content is invigorated by BB aerosols, driven by the ARI effect (Fig. 13b). Through radiation absorption, BB aerosols heat the air, evaporate liquid cloud and promote upward flux of vapor moisture to higher altitude (Fig. 11b), facilitating cloud ice formation there. Similar ice enhancement due to aerosol radiative heating was also seen in simulations of dust-radiation interaction (Dipu et al., 2013). In contrast, the ACI tend to act in opposition to the ARI effect, but at a minor magnitude, showing a moderate negative response. The ACI effect has been reported to invigorate deep convection when more abundant, smaller cloud drops are uplifted to boost the cloud microphysical processes at higher altitude (Rosenfeld et al.,


2008), which, however, is sensitive to the background environment (Khain et al., 2005; Fan et al., 2009). Hints of this effect
are only seen during a narrow time span around 18UTC, and 22 UTC, as indicated by increased cloud ice (Fig. 13a) and
precipitating hydrometeors (Fig. 14a). However, the enhancement is insignificant in magnitude and overwhelmed by the
negative responses that persist during the rest of the diurnal cycle, which may result from different cloud types and
environmental conditions from those in Rosenfeld et al. (2008). Generally, the monthly mean domain averaged results show a
negative effect of the ACI on cloud ice. However, the role of the ACI could be more complicated than found here, because the
ACI effect may potentially modulate the impact of ARI on the cloud ice (Shi et al., 2014; Huang et al., 2019), e.g., by
influencing latent heat release, since the ACI effect is turned on when the ARI effect is assessed. The overall increase in cloud
ice is in agreement with the fine-resolution simulation results over the biomass burning area by the GATOR-GCMOM model
(Ten Hoeve et al., 2012). Although the absolute response of ice concentration is smaller compared with the cloud water change,
the relative change in ice concentration is remarkable (Fig. 12b) (Lee et al., 2017). It should be noted that uncertainties
associated with BB aerosol effects on cloud ice exist, because of the lack of IN parameterization (Fan et al., 2018). Field
observations suggested that the BC in the BB aerosols could contribute substantially to ice nucleation (McCluskey et al., 2014),
which may influence the estimate of the response of cloud ice to BB aerosols.

## 4.4 Impact on precipitation

To investigate the response of precipitation, the diurnal and vertical distribution of domain-averaged changes in
precipitating hydrometeors (sum of rain, snow, and graupel) based on the EMIS1 scenario is presented in Fig. 14. The domain-
averaged rainwater below the freezing level height of about 5 km shows a prominent negative response to the ACI during most
of the daytime. As discussed before, by acting as CCN, the BB aerosols reduce cloud droplet radius, slow down the conversion
rate from cloud to rain, and therefore inhibit warm rain formation. On the other hand, consistent with the responses of cloud
ice, precipitating hydrometeors are enhanced by the ACI effect in the local afternoon and early night due to the invigorated
convection. Generally, an overall reduction in precipitation is induced by the ACI, similar to previous WRF-Chem simulations
of BB aerosol microphysical effects in the Amazon (Wu et al., 2011b). The results of the ARI effect show an overall positive
impact on precipitating hydrometeor concentrations (Fig. 14b) and consequent surface precipitation (Fig. 15a) at the EMIS1
scenario. With the influence of the ARI effect, significant enhancement appears in the precipitating hydrometeors above the
freezing level beginning in the early morning, indicating cold rain processes. Specifically, the graupel concentration, which is
mainly responsible for the cold rain response (Fig. S7) is promoted as more supercooled cloud droplets are present due to the
ARI and efficiently feed the growth of graupel. The increased supercooled cloud water concentration could be a result
originating from the enhanced updraft promoted by the ARI (Fig. 11b). By 17:00–18:00 UTC, the precipitation reaching the
surface is increased correspondingly. The increase in precipitation by the local ARI effect was also found in previous
simulations of light-absorbing aerosols, including black carbon (Lin et al., 2016;) and mineral dust (Dipu et al., 2013; Shi et
al., 2014; Huang et al., 2018). Influenced by the overall effect of both the ACI and ARI mechanisms, a lower concentration of





precipitating hydrometeors is produced by BB aerosols (Fig. 14c) in the morning and afternoon, while enhanced precipitation hydrometeor abundance occurs in a narrow time span from local noon to early afternoon. The variation of convection response throughout the convective evolution cycle implies a possible dependence of aerosol-radiation-cloud interactions on

environmental stabilization, which is also shown by the observation that BB aerosols tend to increase precipitation under unstable conditions (Goncalves et al., 2015).

The response of the precipitation rate to different emission intensities of BB aerosols (represented as AOD) is shown in Fig. 15a. The precipitation reduction by ACI is climatically significant in all emission scenarios, with a monthly mean change of –4% and –19% at EMIS1 and EMIS6, respectively. The precipitation rate is slightly increased by the ARI at low aerosol

loading due to invigorated daytime precipitation as discussed above. However, at high emission intensity, the strong radiative dimming effect of BB aerosols dramatically reduces surface heating (Table 5), which damps the ARI-induced convection invigoration (Fig. S6b) and leads to an overall suppression of convection and a significant reduction of precipitation (Rosenfeld et al., 2008), as reflected by diminished liquid clouds (Fig. S8) and precipitating hydrometeors (Fig. S6b). This dimming effect is even more pronounced than the ACI effect in reducing precipitation for the EMIS6 scenario (Fig. 15). Taking the ACI and

ARI effects together, the monthly mean precipitation rate is decreased by BB aerosols at all emission scenarios used in this study. A reduction of –5% and –23% is calculated for the EMIS1 and EMIS6 emission scenarios, respectively, aligning in magnitude with a precipitation change by –14.5% for the switch of aerosol loading from low emission to high emission scenario in the Amazon found by Thornhill et al. (2018). The precipitation occurrence (calculated as the ratio of precipitating grid cells to the total domain grid cells over the simulation period), which is approximately 11% in the clean case, is reduced noticeably

by both the ACI and ARI effects, indicating more extensive dry area coverage due to the presence of BB aerosols and threatening an aggravation of the precipitation shortage for the Amazon forest in the dry season (Cox et al., 2008).

## 5 Conclusion

In this study, a comprehensive assessment of the impacts of BB aerosols on the regional radiation balance, cloud

properties, and precipitation and their sensitivity to inter-annual variations of BB aerosol emissions was conducted using the fully coupled WRF-Chem model with a 3-km resolution domain in the central Amazon Basin for the dry season. Parallel numerical experiments were performed with different emission scenarios by scaling up and down the original emission rate of the year 2014. These experiments with varying emission scenarios, together with experiments switching off the aerosol-radiation interactions in the model were performed to separate the effects of ARI and ACI, which enables us to quantify each

effect individually and compare their relative significance.

The results show that the shortwave RF by BB aerosols is the outcome of a competition between positive RF by the ARI effect and negative RF by the ACI effect, which is driven largely by the cloud response. The positive shortwave RF associated with cloud reduction due to the semi-direct effect of the BB aerosols counteracts the negative direct shortwave RF and





constitutes the dominant component of ARI-induced effective shortwave RF. Contrarily, the ACI-induced more numerous, but
smaller, cloud radius increase cloud albedo and thereby exert a negative indirect shortwave RF. The relative significance of
the ACI and ARI effects varies with aerosol loading, with a dominant role of the former at low aerosol emission rate while the
latter dominates at high emission intensity. The positive longwave RF by BB aerosols is driven by the ARI effect, through
both aerosol direct radiative forcing and subsequent adjustment of enhanced ice cloud. The all-band aerosol RF is –0.2 W m⁻²
and 1.5 W m⁻² for BB aerosols in the EMIS1 and EMIS6 scenarios, respectively. Surface shortwave radiation is reduced by
BB aerosols, with an estimate of –17.1 W m⁻² for a multi-year averaged emission intensity (EMIS3), which is compensated
mostly by suppression of sensible and latent heat flux from ground to the atmosphere. The response of cloud LWP to BB
aerosols is driven in opposite directions by the ARI and ACI effects. The surface cooling generated by radiation extinction
together with the atmospheric heating from absorption of solar radiation stabilizes the atmosphere, inhibits convection
development, and thereby decreases the cloud LWP. In contrast, higher cloud LWP is produced by the ACI through inhibited
warm rain formation. The relative significance of the competing effects depends on the aerosol amount, consistent with the
aerosol shortwave radiative forcing response, implying a crucial role of cloud adjustments in determining aerosol radiative
forcing on the Earth-atmosphere system. Enhanced cloud IWP with the presence of BB aerosols is related to a stronger upward
flux of water vapor induced by the ARI effect.

Lower precipitation occurrence is induced by both the ARI and ACI effects, which implies a larger fraction of dry areas
in the Amazon Basin when affected by BB aerosols, threatening to exacerbate droughts during the dry season. The domain-
averaged precipitation rate is diminished substantially by ACI consistently over all the emission scenarios used in this study,
implying the importance of including ACI effects on the sub-grid cumulus convection when applying large-scale simulations
at coarse grid resolution (Archer-Nicholls et al., 2016). Strong suppression of warm rain formation is responsible for the
precipitation reduction caused by the ACI, but in the lower emission scenarios, an ACI-induced invigoration of deep convection
occurs during a narrow time period, due to latent heat release from more abundant smaller droplets aloft (Rosenfeld et al.,
2008). The precipitation response to the ARI effect is nonlinear due to the effects of mixed-phase precipitation. At low BB
aerosol emission rates, enhanced mixed-phase precipitation is found as a result of higher graupel content with the enhanced
supply of supercooled cloud droplets by the ARI, while the invigoration disappears in the high emission scenarios with reduced
presence of supercooled cloud droplets due to overwhelming suppression of convection by BB aerosols. Reduction in monthly
mean precipitation rate by the overall effects of BB aerosols is found for all emission scenarios, and intensifies with aerosol
loading, which may imply a positive feedback between precipitation scavenging and aerosol concentration for intense BB
events. A reduction of monthly mean precipitation rate by –5% and –23% is estimated for the EMIS1 and EMIS6 scenarios,
respectively, suggesting a strong sensitivity of precipitation to aerosol concentration.

The high sensitivity and nonlinear relationship between regional radiation, liquid water content, precipitation, and BB
aerosol abundance highlight the importance of comprehensive assessments of BB aerosol effects in the Amazon with
multiannual aerosol emission scenarios. The variation of the ACI and ARI effect with increasing aerosol emission revealed a



saturated tendency for the ACI while a continuing increasing role of the ARI at high aerosol loading. This may shed light on the climatic importance of the ARI at highly polluted regions and episodes with severe combustion aerosol emissions such as intensive wildfires, industrialization-related fossil fuel combustion, and agricultural crop waste burning. The key role of the

ARI also highlights the importance of accurate representation of aerosols and their optical properties in models when addressing their climate effects.

It should be noted that this study only focuses on the local effects of BB aerosols for a typical region in the Amazon Basin. The large-scale response in the atmospheric field (Lee et al., 2014) caused by horizontally inhomogeneous responses to unevenly distributed aerosols is out of the scope of this study. The role of aerosols acting as IN has not been included in the

WRF-Chem model used here. Parameterization of this mechanism is needed to better quantify aerosol effects on climate. In addition, further investigations on the formation mechanisms and light absorption associated with SOA are needed to better parameterize the physical and optical properties of organic aerosols in the model (Shrivastava et al., 2017, 2019), in order to better recognize the role of BB aerosols in the climate system. Furthermore, the sensitivity of the climate response to the concentration of BB aerosols may be influenced by the meteorological conditions, and as this study is based on September

2014, continuing investigation is needed to characterize the influence of variations in meteorological factors.

**Author contributions**

Y.C. and H.S. designed and led the study. L.L. performed the model simulation and analyzed the data. L.L., H.S. and Y.C. interpreted the results. M.O.A., M.S. and U.P. discussed the results. M.P. and C.P. contributed data for model validation. L.L. wrote the manuscript with input from all coauthors.

**Competing interests**

The authors declare that they have no conflict of interest.

**Acknowledgements**

We thank the Max Planck Society (MPG) for support and Minerva Program of MPG. We acknowledge the MODIS, TRMM, AERONET and IGRA teams for the data used in this study. Furthermore, we would like to thank for the support by

the ATTO project and all the people involved in it. We would like to acknowledge the German Federal Ministry of Education and Research (BMBF contracts 01LB1001A and 01LK1602B), supporting this project as well as the construction and operation of the ATTO site. We also acknowledge the support of the Brazilian Ministério da Ciência, Tecnologia e Inovação (MCTI/FINEP contract 01.11.01248.00) as well as the Amazon State University (UEA), FAPEAM, LBA/INPA and SDS/CEUC/RDS-Uatumã for their support during construction and operation of the ATTO site. Dr. Shrivastava was supported



by the U.S. Department of Energy (DOE), Office of Science, Office of Biological and Environmental Research through the
       Atmospheric System Research (ASR) and Early Career Research Programs.

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



**Table 1.** WRF-Chem configuration.

| Atmospheric Process | WRF-Chem Option |
|---|---|
| Longwave radiation | RRTMG |
| Shortwave radiation | RRTMG |
| Surface layer | Monin-Obukov |
| Land surface | RUC |
| Boundary layer | YSU |
| Microphysics | Lin et al. |
| Cumulus | Grell-Devenyi ensemble scheme in the 75 km and 15 km simulations; no cumulus scheme in the 3 km simulation |
| Gas-phase chemistry | CBMZ |
| Aerosol chemistry | MOSAIC |
| Aqueous-phase chemistry | Fahey and Pandis |
| Photolysis | Fast-J |
| Anthropogenic emissions | EDGAR-HTAPv2 |
| Biogenic emissions | MEGAN |
| Biomass burning emissions | FINNv1.5 |

960





**Table 2.** Experiment design description.

| Experiment identification | Experiment description |
| --- | --- |
| CC3 | Clean case at 3 km resolution without BB emission. |
| CCNR3 | Clean case at 3 km resolution without BB emission. The aerosol radiation feedback is turned off. |
| PC3_EMISX | Polluted case at 3 km resolution with BB emission scenario EMISX. EMISX represents scenario with BB aerosol emission rate scaled by a factor of X based on original BB emission. |
| PCNR3_EMISX | Polluted case at 3 km resolution with BB emission scenario EMISX. The aerosol radiation feedback is turned off. |





**Table 3.** Statistical indexes of the comparisons between modeled and observed surface air temperature (T), relative humidity (RH) and wind speed (WS) at the ATTO site over September 2014.

|  | MB | RMSE | r |
|---|---|---|---|
| T (°C) | 0.2 | 1.5 | 0.86 |
| RH (%) | −2.3 | 9.2 | 0.78 |
| WS (m s$^{-1}$) | −0.2 | 1.9 | 0.52 |

MB: the mean bias;
RMSE: the root mean square error;
r: the correlation coefficient.





**Table 4.** Comparison of SSA at 550 nm obtained from model simulation and observation.

|  | Observation | Model[a] |
| --- | --- | --- |
| Alta_Floresta (AERONET) | 0.95±0.00 (average of Sep 2014) | 0.88±0.01 |
| Ji_Paraná_SE (AERONET) | 0.95±0.01 (average of Sep 2014) | 0.88±0.01 |
| Porto Velho (Reid et al., 1998) | 0.86±0.05 (average of 5-13 Sep 1995) | 0.88±0.01 |
| TT34[b] (Rizzo et al., 2013) | 0.87±0.06 (average of Jul–Dec 2008–2010) | 0.90±0.01 |
| ATTO[c] (Saturno et al., 2018) | 0.88 (average of Aug–Nov 2012–2017) | 0.90±0.01 |

a) Model results with EMIS1, averaged for September 2014.
b) The SSA values at this site are for 637 nm. Calculation of SSA at 550 nm is not conducted due to incomplete information on Angstrom exponent in Rizzo et al. (2013).
c) The SSA observation for the ATTO site is obtained from Saturno et al. (2018b) by extrapolating the original value at 637 nm to that at 500 nm using the Angstrom exponents in Saturno et al. (2018b).





**Table 5.** Summary of monthly mean perturbations caused by ARI, ACI and the overall effect from BB aerosols averaged over domain3 for the EMIS1 and EMIS6 emission scenarios.

| Variable | ARI | | ACI | |
|---|---|---|---|---|
| | EMIS1 | EMIS6 | EMIS1 | EMIS6 |
| TOA solar radiation (W m$^{-2}$) | 0.4 | 2.0 | –0.7 | –1.4 |
| TOA solar + IR radiation (W m$^{-2}$) | 0.5 | 3.0 | –0.7 | –1.5 |
| Surface solar radiation (W m$^{-2}$) | –5.7 | –30.5 | –0.6 | –1.3 |
| Sensible heat flux (W m$^{-2}$) | –2.3 | –14.4 | –0.1 | –0.2 |
| Latent heat flux (W m$^{-2}$) | –2.0 | –11.8 | –0.5 | –1.1 |
| Surface temperature (°C) | –0.03 | –0.20 | +0.00 | +0.01 |
| PBL height (m) | –8 | –58 | +0 | 2 |
| Cloud droplets number (cm$^{-2}$) | –0.7×10$^5$ | –6.0×10$^5$ | 4.0×10$^5$ | 14.5×10$^5$ |
| Cloud droplets radius (μm) | –0.7 | –0.5 | –1.0 | –2.6 |
| Cloud base height (m) | –6 | –40 | 1 | 5 |
| LWP (g m$^{-2}$) | –0.6 | –3.8 | 0.7 | 1.7 |
| LWP in PBL (g m$^{-2}$) | 0.03 | 0.14 | –0.01 | –0.04 |
| LWP in FT (g m$^{-2}$) | –0.6 | –3.9 | 0.7 | 1.7 |
| IWP (g m$^{-2}$) | 0.04 | 0.26 | –0.02 | –0.07 |
| Precipitation (mm day$^{-1}$) | 0.01 | –0.11 | –0.06 | –0.10 |



**Figure 1.** Model domain and orography. The outer map represents the parent domain with 75 km horizontal grid spacing, and the embedded squares show the extents of the 15-km (d02) and 3-km (d03) nested domains. Red dots denote AERONET monitoring stations; the triangle represents the ATTO site.

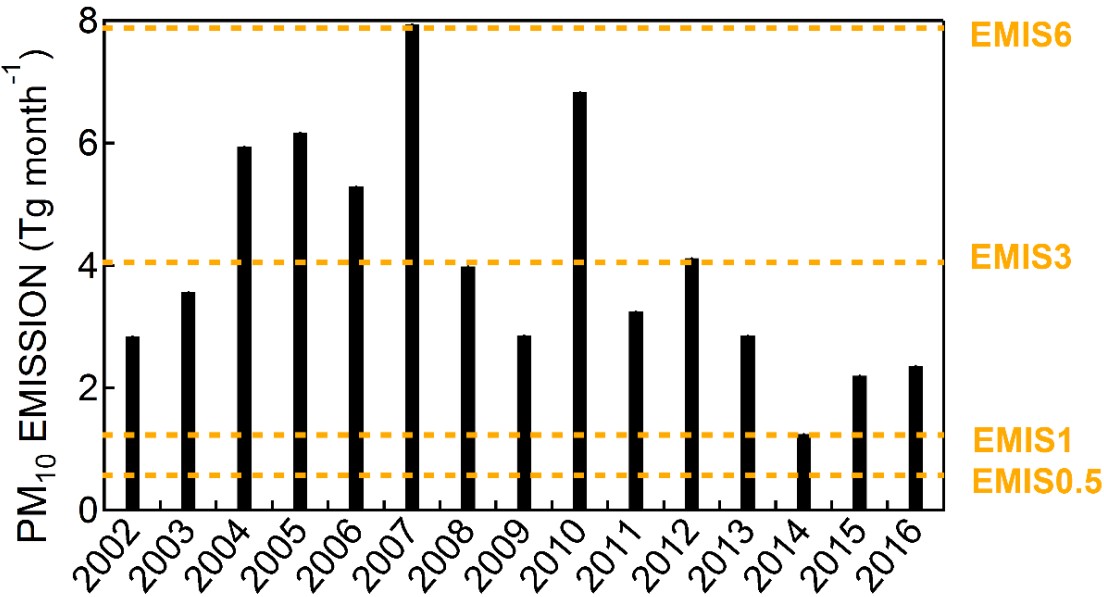

**Figure 2.** Annual variation of PM$_{10}$ emission during September over domain1 based on FINNv1.5.



**Figure 3.** Time series of surface air temperature (a), relative humidity (b) and wind speed (c) from the domain3 simulation and the observations at ATTO during September 2014.





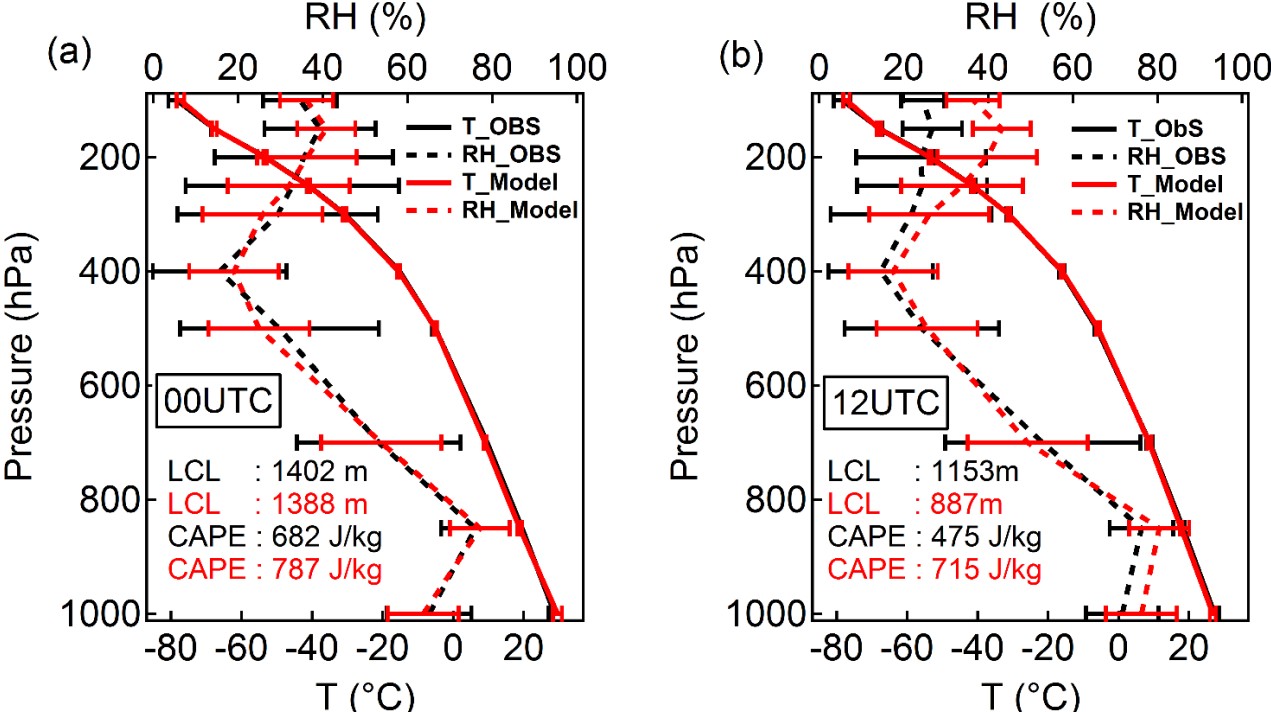

**Figure 4.** Vertical profiles of air temperature, relative humidity at standard levels, and retrieved CAPE and LCL values from radiosonde observations and WRF-Chem domain3 simulations at 00:00 UTC and 12:00 UTC at Manaus. Error bars at each pressure level represent the standard error at that level.



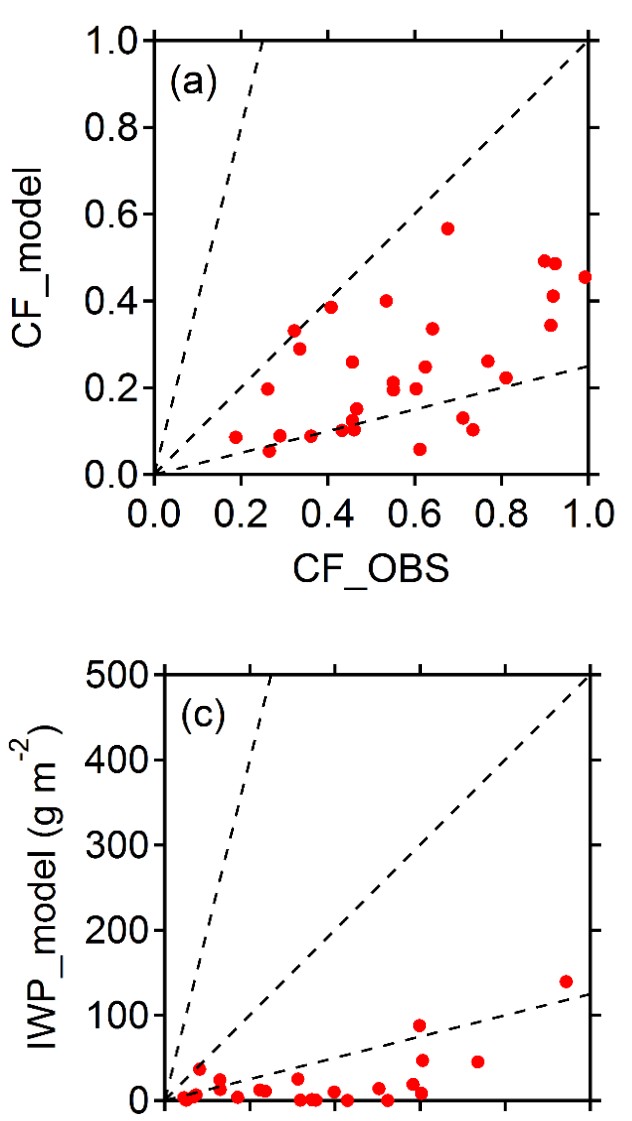

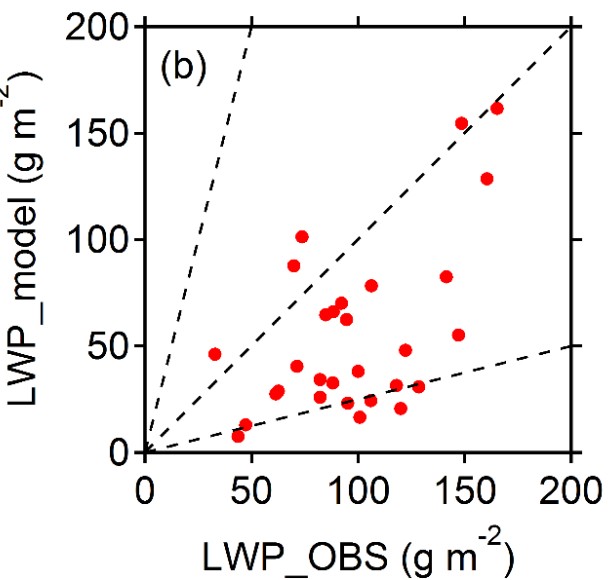

**Figure 5.** Scatter plots of cloud fraction (a), total liquid water path (b) and total ice water path (c) from WRF-Chem domain3 simulations and MODIS satellite measurements.





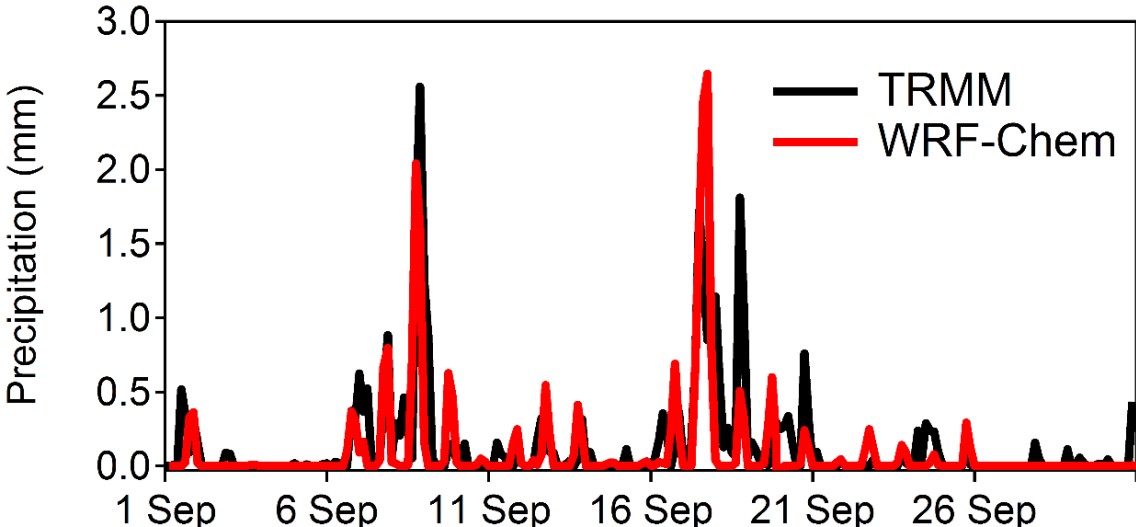

**Figure 6.** Time series of region averaged 3-hour accumulated precipitation (mm) over domain3 from TRMM satellite observations and WRF-Chem simulations during September 2014.





**Figure 7.** Time series of AOD at 550 nm from AERONET measurements, MODIS daily retrievals, and WRF-Chem simulations for three AERONET stations. The locations of the AERONET stations are shown in Figure 1.

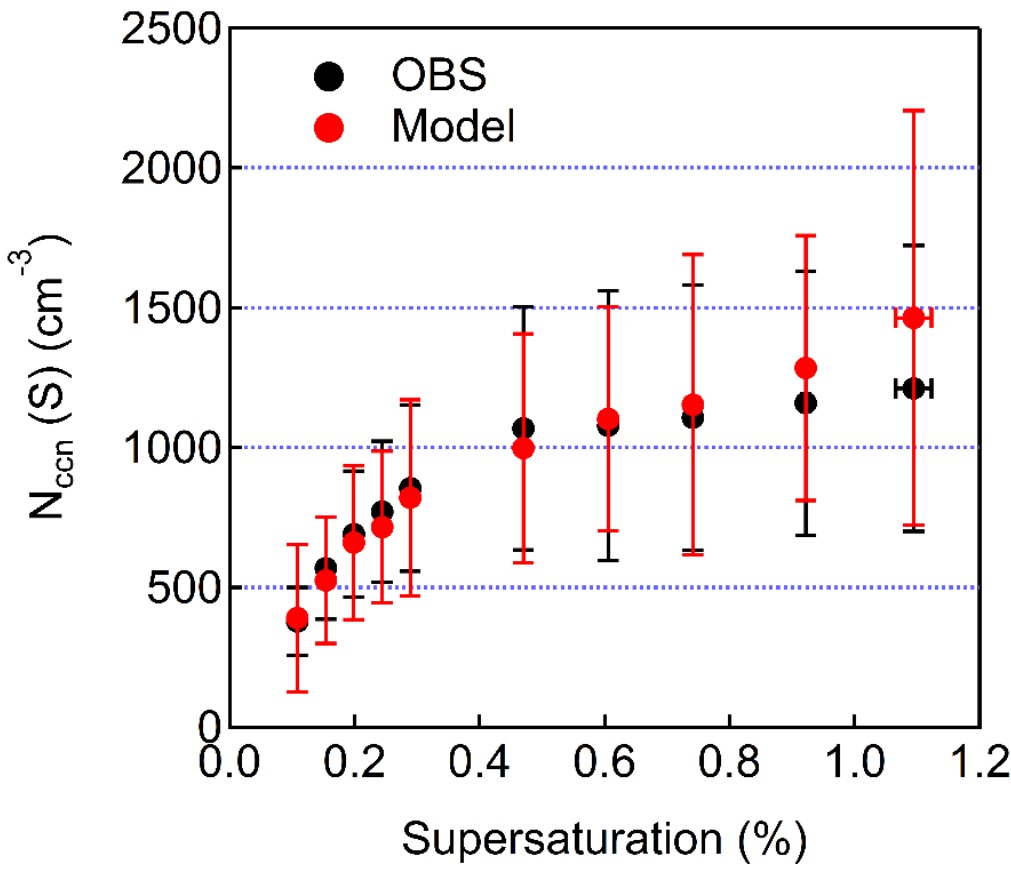

**Figure 8.** Monthly averaged CCN number concentrations at different supersaturations from ATTO observations and WRF-Chem simulations. Error bars represent the standard deviation.



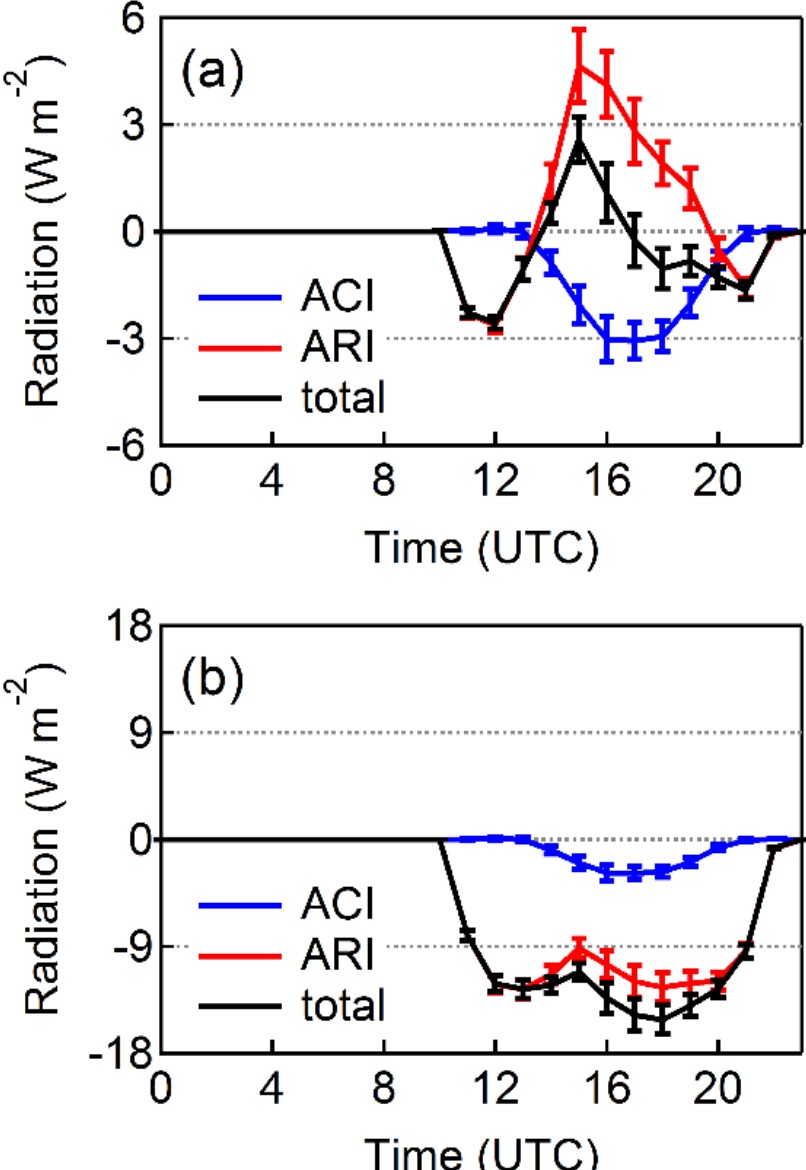

**Figure 9.** Diurnal variation of changes in all-sky shortwave radiation at TOA (a) and surface (b) in the EMIS1 emission scenario. Error bars denote the standard error.





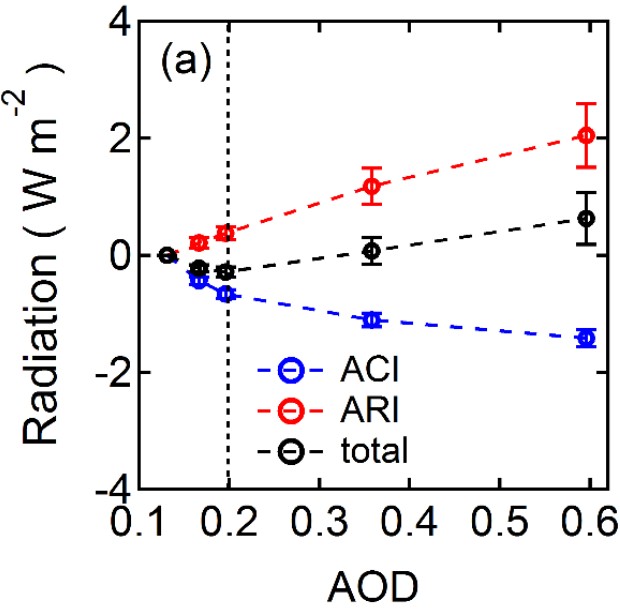

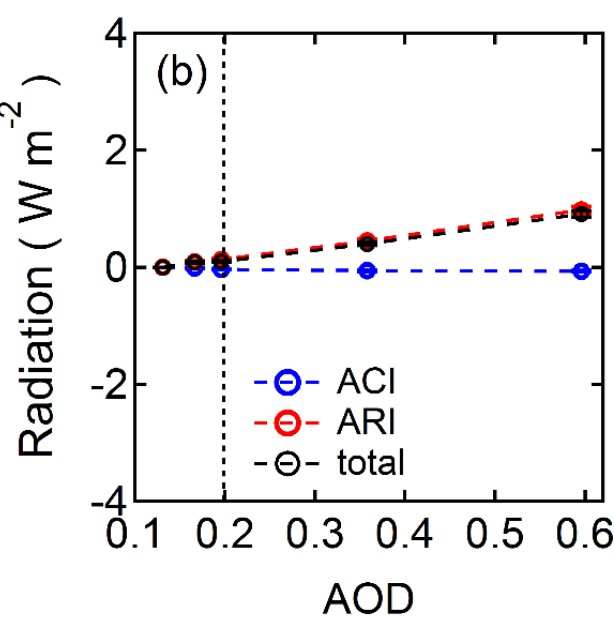

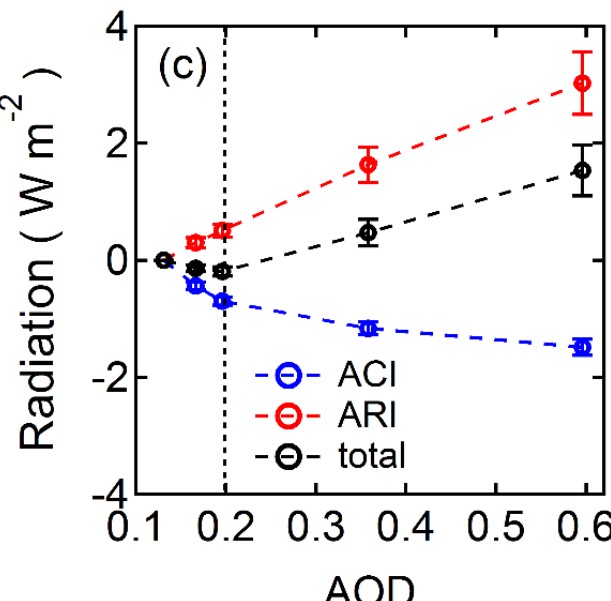

**Figure 10.** Changes in shortwave (a), longwave (b), and total (c) radiation budgets at TOA with increasing BB emission intensity (indicated by domain-averaged AOD for each emission scenario). The vertical dotted line in each plot indicates the EMIS1 scenario. Error bars denote the standard error.





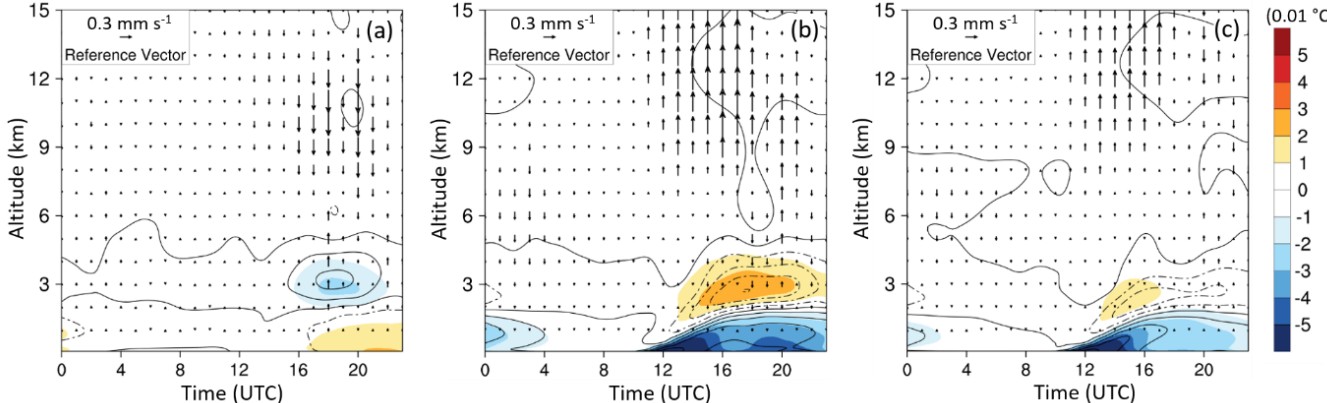

**Figure 11.** Domain-averaged difference in air temperature (shaded, in intervals of 0.01°C), relative humidity (contour lines, in intervals of 0.1%), and updraft velocity (arrows) caused by BB aerosols' ACI (a), ARI (b), and total effect (c) in the EMIS1 emission scenario.

**Figure 12.** Changes in cloud LWP (a) and cloud IWP (b) with increasing BB emission intensity (indicated by domain-averaged AOD in each emission scenario). The vertical dotted line in each plot indicates the EMIS1 scenario. Error bars denote the standard error.





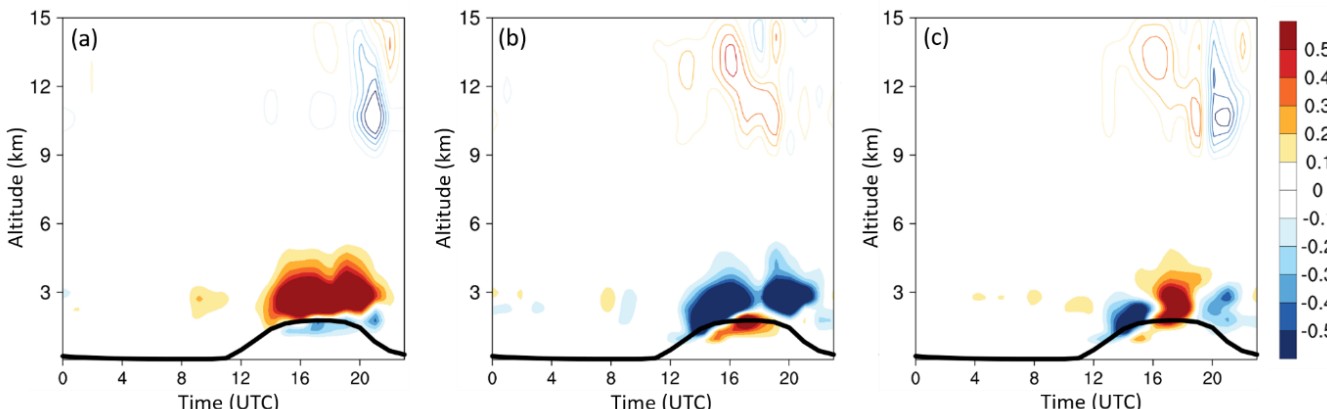

**Figure 13.** Diurnal variation of the vertical distribution of the domain-averaged difference in cloud water (shaded, in mg kg$^{-1}$) and cloud ice (contour lines, in 0.1 mg kg$^{-1}$) caused by BB aerosols' ACI (a), ARI (b), and total effect (c) in the EMIS1 emission scenario. The thick black line represents the PBL height.





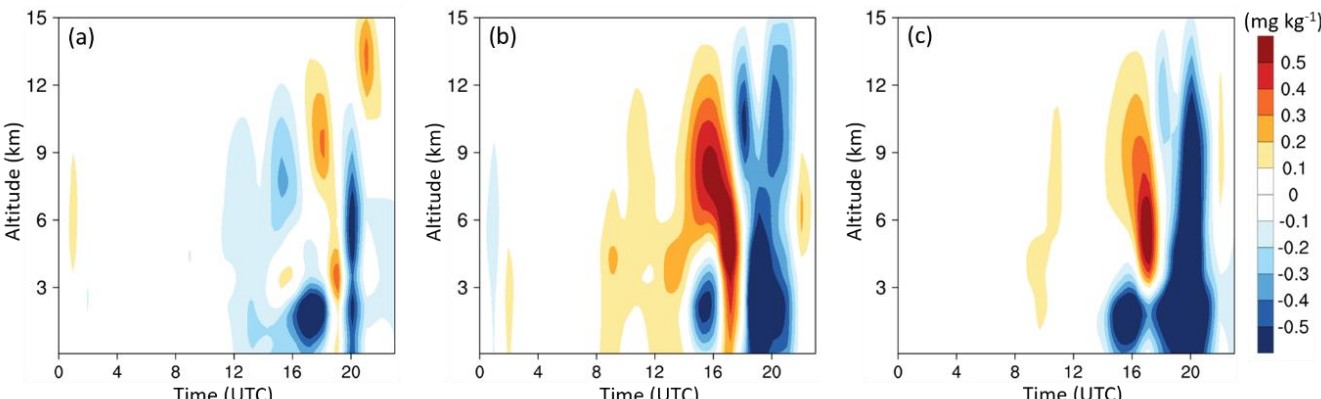

**Figure 14.** Diurnal variation of the vertical distribution of the domain-averaged difference in precipitating hydrometeor (QRAIN+QSNOW+QGRAUP) concentrations caused by BB aerosols' ACI (a), ARI (b), and total effect (c) in the EMIS1 emission scenario.

**Figure 15.** Changes in domain-averaged precipitation rate (a) and precipitation occurrence (b) with increasing BB emission intensity (indicated by domain-averaged AOD in each emission scenario). The vertical dotted line in each plot indicates the EMIS1 scenario. Error bars denote the standard error.