# Peer review of "Impact of biomass burning aerosols on radiation, clouds, and precipitation over the Amazon during the dry season: dependence of aerosol-cloud and aerosol-radiation interactions on aerosol loading"

_Atmospheric Chemistry and Physics, 2020_

## Referee Comment (RC1) · Anonymous Referee #1 · 18 May 2020

This research aims to quantify the impact of biomass burning on radiative forcing in Amazon forest. The authors present a very comprehensive analysis and I believe that this research would help us better understand the climate impact of biomass burning in this region. Improvements are needed to make the manuscript scientifically sounding and suitable for publishing, as follows:

Major concerns:

1. In Section 3, was the WRF simulation nudged towards FNL analysis data? If so, is it fair to compare nudged model results with observations? Please clarify on this.

2. Is the conclusion sensitive to different plume rise parameterizations? Can the authors provide some validations on the simulated plume rise heights?

3. Is the conclusion sensitive to the choice of the period? One month seems to be quite short. Why not consider multi-month or multi-year analysis?

4. The main conclusion is that lower precipitation is expected with biomass burning aerosols. Do historical observations support this conclusion?

5. Why did an underestimation of precipitation during Sept 17 & 18 lead to much lower temperature and higher RH, compared to the observations? Why did the temperature vary little these days? Maybe it would be worthy to check the synoptic pattern and assure that this is not a model bug.

6. The validations of AOD simulation are not so impressive, it would be helpful to show: 1. the mean absolute bias & correlation between simulation and observation, making the validation more quantitative; 2. reference other papers for the bias between simulated and observed AOD in South America and other regions, quantitatively.

7. The authors may consider to move Section 3 to supplement to make the manuscript less length and more focused.

Technical comments:

Line 27: 'which enables them' -> 'which enable them'

Line 151: Need a reference

---

## Referee Comment (RC2) · Anonymous Referee #2 · 25 May 2020

**Review of "*Impact of biomass burning aerosols on radiation, clouds, and precipitation over the Amazon during the dry season: dependence of aerosol-cloud and aerosol-radiation interactions on aerosol loading*" authored by Liu et al.**

The manuscript presents a comprehensive study of biomass burning (BB) aerosol effects on radiation, clouds, and precipitation over the Amazon, addressing both aerosol-cloud and aerosol-radiation interactions (ACI and ARI). The cloud-resolving WRF-Chem model and a suite of ground-based and satellite observations over the Amazon were employed. It is great to see that substantial efforts have been made to evaluate the model's performance in various aspects. The major findings about the distinctive effects of ACI and ARI based on the sensitivity analyses reinforce the importance of equally considering those two effects in future aerosol effect assessments. The paper is well written overall. I would recommend the publication of this study by ACP, after the following issues are addressed.

1) To assess ARI, why not contrasting the experiment PC3_EMISX and PCNR3_EMISX? The current way to obtain ARI has an underlying assumption that that the total aerosol effects are a linear combination of ACI and ARI, which may not be the case because of the complexity of the nonlinear microphysics-dynamics-thermodynamics interactions of the system. Such an uncertainty should be discussed in the paper.

2) It is unclear how the model treats the BC aging process. According to the present model description in Section 2.1, it seems the fresh BC are immediately mixed with other types of aerosols after emission. Such a simplified treatment could result in overestimation of the BC absorption and associated radiative forcing [Wang et al., 2018; Peng et al., 2016].

3) According to Fig. 6, the month-long simulations include a couple of deep convective systems with heavy precipitation (Sept. 9, 17-18). For the precipitation response analyses in Fig. 15, can the authors take a further step to assess the deep convective systems and the rest separately? Maybe a threshold of 3 mm/3hr can be applied to categorize those cases.

4) For the IWP evaluation, how are the satellite data are averaged spatially? It seems the satellite observations shown in Fig. 5 are averaged over cloud points only. I doubt ice heterogeneous nucleation scheme can explain such a huge discrepancy. Even if the ice production scheme is not a function of INP concentration in this microphysics, it should still exist (most of time as function of temperature).

5) Line 62-65, similarly, a recent study using satellite data shows nonlinear response of deep convective clouds to smoke aerosol in South America [Jiang et al., 2017].

6) Out of personal curiosity, to what extent the FDDA can reduce the meteorological biases? If the authors have the model free run available, I like to see a comparison of those two.

7) Fig. 4, what are the two dash lines in addition to the 1-to-1 line?

8) Line 326-327, it doesn't make much sense to compare a regional aerosol forcing to the global values.

9) Fig. 11 is discussed after Figs. 12 and 13. Better to reverse their order.

10) In Fig. 11 and 12, the larger updraft velocity and IWP by absorbing aerosols corroborate the thermodynamic invigoration hypothesis by Wang et al. [2013] which suggested larger CAPE above PBL due to the presence of absorbing aerosols in the lower troposphere.

11) Title is too long. Maybe remove *"dependence of aerosol-cloud and aerosol-radiation interactions on aerosol loading"*.

References:

- Jiang, J., H. Su, L. Huang, Y. Wang, S. Massie, B. Zhao, Z. Wang, and O. Ali, "Contrasting Effects on Deep Convective Clouds by Different Types of Aerosols", Nat. Communications 9(3874), (2018)
- Peng, J., M. Hu, S. Guo, Z. Du, J. Zheng, D. Shang, M. Zamora, L. Zeng, M. Shao, Y. Wu, J. Zheng, Y. Wang, C. Glen, D. Collins, M.J. Molina, R. Zhang, "Markedly enhanced direct radiative forcing of black carbon particles under polluted urban environments", Proc. Natl Acad. Sci. USA, 113(16), 4266-4271 (2016)
- Wang, Y., P. Ma, J. Peng, R. Zhang, J.H. Jiang, R. Easter and Y. Yung, "Constraining Aging Processes of Black Carbon in the Community Atmosphere Model Using Environmental Chamber Measurements", J. Adv. Model. Earth Syst. 10(10), 2514-2526 (2018)

---

## Author Response (AR1)

**Response to Comments of Referee#1**

Dear Reviewer:

We would like to thank you for the valuable and constructive comments/suggestions which helped to improve our manuscript. We have carefully revised the manuscript accordingly. Please find our point-to-point responses below (line numbers and figure numbers refer to the new version of manuscript; reviewer comments and suggestions are in italics, responses are in plain font; revised sections in the manuscript text in response to the comments are marked with red color).

*1. In Section 3, was the WRF simulation nudged towards FNL analysis data? If so, is it fair to compare nudged model results with observations? Please clarify on this.*

**Response:** We thank the reviewer for pointing out this missing information. In this study, a nudging towards FNL analysis data was used for the domain1 (75 km) and domain2 (15 km) to provide a more accurate meteorological boundary for domain3 (3 km). The innermost domain was driven one-way by initial and boundary inputs from the outer domain. No nudging was used in the domain3 (3 km). To evaluate the impact of aerosol-induced perturbations, we used only the meteorological fields in domain 3.

We have added clarifications on this point (Page 6, line 156): 'No nudging was used in the innermost domain. The aerosol-induced perturbations were estimated with the meteorological fields simulated in domain3.'

*2. Is the conclusion sensitive to different plume rise parameterizations? Can the authors provide some validations on the simulated plume rise heights?*

**Response:** Thanks for drawing attention to this point. The plume rise height is indeed an important factor for investigating aerosol-radiation-cloud interaction. For example, Johnson et al. (2004) showed the vertical distribution of light-absorbing aerosols could significantly affect the aerosol radiative effect. The plume rise parameterization used in this study is the only option provided in WRF-Chem (Grell et al., 2011; Freitas et al., 2007), which has shown commendable performance in simulating the plume height in the Amazon (Wu et al., 2011). To demonstrate the performance of the plume rise scheme in our study, we have conducted a comparison of the aerosol vertical distribution using CALIPSO observations.

The clear-sky monthly mean aerosol extinction profiles at 532 nm, provided by the CALIPSO Level 3 aerosol product (Tackett et al., 2018), was used here. The simulation data

was processed to align with the observation by using outputs corresponding to the passing time of the satellite, excluding cloudy grid cells with a cloud criterion of 1 g/kg and interpolating the extinction coefficient at 550 nm to 532 nm. The averaged aerosol extinction profiles from the model results of domain 3 and observation were then compared (Fig. R1a).

The observed vertical profile of the aerosol extinction coefficient is basically reproduced by the model. The model well captures the peaks of the aerosol extinction coefficient at the surface and near 2 km. Similar to our study, an overestimation of the aerosol extinction coefficient above 3 km was also simulated by Wu et al. (2011, Fig. R1b). The phenomenon found in both studies may be caused by an overestimated exchange between PBL and the free atmosphere by turbulent mixing and convective transport, an underestimation of precipitation scavenging, and/or an overestimated plume rise at some fire spots. Generally, the vertical distribution of aerosols is captured reasonably by the model, which illustrates an acceptable performance of the plume rise parameterization and indicates the reliability of the conclusions about aerosol-radiation-cloud interactions obtained by the model.

We have added the comparison of the aerosol extinction coefficient profile to the aerosol evaluation section in the SI (Page 6, Line 142).

[Figure]

(a)                                            (b)

**Figure R1.** (a) Monthly mean clear-sky aerosol extinction coefficient at 532 nm averaged over domain3. (b) Clear-sky aerosol extinction coefficient at 532 nm averaged over the Amazon Basin (Figure 3 from Wu et al. 2011).

*3. Is the conclusion sensitive to the choice of the period? One month seems to be quite short. Why not consider multi-month or multi-year analysis?*

**Response:** We acknowledge the reviewer's concern and we agree that longer term simulation (multi-month and/or multi-year) may add to the robustness of the conclusions. Currently, the simulations for this study were conducted for one month but not for a longer time mainly because of the limitation of the computing resources. As the WRF-Chem model itself is computation-demanding due to many coupled modules, and since additionally the simulations included 3-nested domains with fine resolution (3 km) at the innermost domain of 161*161 grids, and a set of 10 parallel sensitivity cases for 5 emission scenarios, the simulations are highly computational intensive. Previous WRF-Chem simulations using such fine resolution to estimate the aerosol-radiation-cloud interaction were limited to 3-day to 8-day periods (Archer-Nicholls et al., 2016; Wu et al., 2011). The authors made an attempt in this study to expand the simulation time in order to include more cloud and precipitation cases and to give more robust constraints on the aerosol effect assessments compared to previous simulations. Limited by current computing resources, we chose a simulation period of 1 month, which has been shown to be a timescale that has short-term climatic significance (Becker et al., 2013).

The one-month simulation period targets September 2014 in order to make sure the study is conducted under typical dry season conditions. Pöhlker et al. (2016) comprehensively compared the meteorology over 18 years (1998-2016) and found that 2014 is a typical year and September is a typical month of the dry season for the Amazon area, e.g., the precipitation rates on September 2014 were found comparable to the 18-year average data and showed no pronounced hydrological anomalies (Fig. R2). Hence, this sensitivity study based on September 2014 can serve to reveal the typical sensitivity behavior of the dry season radiation, cloud, and precipitation over the central Amazon to BB aerosols. To address the questions of the representativeness of the simulation period and sensitivity of the results to other dry season periods, we added a statement on Line 91 and included a longer period simulation as a suggestion for future investigation on Line 460, respectively:

'Comparison of the precipitation in central Amazonia in the year 2014 with that averaged over 18 years (1998–2016) indicates that the atmospheric conditions in this region in 2014 are climatically representative (Pöhlker et al., 2016). Therefore, the present study based on September 2014 may serve to represent the typical sensitivity behavior of the dry season climate to BB aerosol concentration variations.'

'Furthermore, the sensitivity of the climate response to the concentration of BB aerosols may be influenced by the meteorological conditions, and as this study is based on September 2014, continuing model investigations based on varying and longer periods are needed to characterize the influence of variations in meteorology and to provide climatic assessments.'

[Figure]

Figure R2. Precipitation rates from tropical rainfall measuring mission (TRMM) PTRMM and in situ measurements at the ATTO site PATTO. The $P_{TRMM}$ seasonal cycles are derived from an area upwind of the ATTO site (59.5 W, 2.4 N, 54.0 W, 3.5 S), covering a long-term period from 1 January 1998 to 30 June 2016 (aqua shading), and the period of the CCN measurements from 1 March 2014 to 28 February 2015 (blue line). (Figure 1 from Pöhlker et al., 2016)

*4. The main conclusion is that lower precipitation is expected with biomass burning aerosols. Do historical observations support this conclusion?*

**Response:** The authors appreciate this insightful comment. Practically, the inconsistent methodologies used in observational data analysis and model simulation make it difficult to reconcile observation and model results. The modelling study uses strict control experiments by increasing/decreasing aerosol concentration on a fixed meteorology field. In contrast, satellite and in-situ observations usually use a method of sampling spatial and/or temporal correlation between aerosol and precipitation, and therefore other factors such as concurrent meteorological influence may bring in uncertainties. Besides, mismatched domain, study period, and convection stage between observation and model, e.g., that satellites usually measure at a specific time of day, should also be considered. Observational studies on the impact of biomass burning aerosols on precipitation, especially in-situ observations, are still scarce. The authors did find satellite observations in Rosenfeld et al. (1999) and Koren et al. (2012) showing overall consistent results with the conclusions in this study. Satellite measurements of a biomass burning episode in the tropical area by Rosenfeld et al. (1999) found that the rain formation process was shut down by biomass burning aerosols. This is consistent with our model results that biomass burning aerosols cause decreased precipitation occurrence and consequently lower precipitation amount. Another satellite measurement over the Amazon from June to August 2007 by Koren et al. (2012) showed a trend of decreasing precipitation due to biomass burning aerosols at an aerosol loading similar to this study (AOD >0.2). Yet, due to uncertainties caused by uncoordinated observation measurement and model simulation, the authors would be cautious and conservative about directly comparing the observations with the modelling results. More

observation measurements and observation conducted in coordination with modelling might still be needed to provide constraints on aerosol-precipitation interactions in the model.

*5. Why did an underestimation of precipitation during Sept 17 & 18 lead to much lower temperature and higher RH, compared to the observations? Why did the temperature vary little these days? Maybe it would be worthy to check the synoptic pattern and assure that this is not a model bug.*

**Response:** Compared to the observations, the model underestimated the precipitation at the ATTO site during Sept 17 & 18 (Fig. S3), and meanwhile the temperature (Fig. S2a) in the model (red line) is higher than the observations (dot) and the RH (Fig. S2b) in the model (red line) is lower than the observations (dot). This may be associated with the fact that the precipitation itself moistens the soil, enhances the latent heat flux, and therefore leaves less net energy in the ground to heat the surface air, so low surface air temperature occurs during precipitation. Hence, the underestimation of precipitation in the model was accompanied by higher modelled temperature. As the RH is inversely related to air temperature, a lower modelled RH was found when precipitation was underestimated. We have added more explanation about the relationship between the bias in precipitation and in temperature and RH (Line 78 in SI): 'This is expected since precipitation enhances surface evaporation and latent heat flux, leaves less net energy in the ground to heat the surface air (Zhuang et al., 2017), and therefore corresponds to a cool and moistened near-surface atmospheric state as can be found in the ATTO observation (Fig. S2). Consequently, the precipitation underestimation in the model was accompanied by higher biased simulated temperature and lower biased RH.'

The discrepancy of precipitation, temperature, and RH on Sept 17 & 18 between model and observation is obvious at the ATTO site but is not evident for the domain averaged precipitation (Fig. S6, comparison between simulated domain precipitation and TRMM observation). As suggested by the reviewer, a check of the synoptic pattern against the NCEP reanalysis was conducted (Fig. R3, shown below) and it shows the model basically captured the synoptic patterns. This means the model simulated the synoptic patterns and regional precipitation well but underestimated the precipitation at the ATTO site. The reason for this precipitation underestimation at the ATTO site could be associated with a location bias of rainfall. We then extracted the air temperature and RH from the grid (north of ATTO) where significant precipitation occurred on Sept 17&18 and compared it with the ATTO observations (Fig. R4, shown below). The air temperature (RH) at the precipitating grid dropped (increased) comparably to the observation during the rainfall period. It also serves as an indicator that the location bias of the rainfall on Sept 17 & 18 could account for the discrepancy of the meteorological factors between the simulation and the ATTO observation. We have added

more explanation for the underestimation of precipitation at ATTO (Line 113 in SI): 'The regional rainfall events on 6, 8, 17 and 18 Sep are well predicted by the model with a slight underestimation, which reflects a better model performance compared with the evident model underestimation of precipitation at the ATTO site on these four days. Moreover, the modelled synoptic patterns corresponding to the precipitation episodes are consistent with the NCEP reanalysis data (not shown). The well-reproduced regional synoptic and precipitation conditions in the model serve to corroborate that the precipitation underestimation at the ATTO site is likely induced by a local bias of rainfall location and neglecting precipitation of sub-grid convection by the model.'

[Figure]

**Figure R3.** Surface air pressure and horizontal wind from NCEP reanalysis and WRF-Chem model.

[Figure]

**Figure R4.** Time series of surface air temperature (a), relative humidity (b) from ATTO observation (black dot) and the domain3 simulation at ATTO (red line) and at significant precipitaiton point (blue line) during September 2014.

*6. The validations of AOD simulation are not so impressive, it would be helpful to show: 1. the mean absolute bias & correlation between simulation and observation, making the validation more quantitative; 2. Reference other papers for the bias between simulated and observed AOD in South America and other regions, quantitatively.*

10 **Response:** Thanks for the suggestion. We have revised the AOD evaluation in the studied domain according to the reviewer's suggestion to make it more clear and quantitative (Line 123 in SI).

'Table S3 shows the comparison of the modelled AOD against the AERONET observation at Manaus_EMBRAPA, a forest reservation site representative of the central Amazon environment (Artaxo et al., 2013). The model simulation generally captures the absolute value and the temporal variation of the observed AOD, with the mean bias and correlation coefficient being -0.03 and 0.54, respectively (Table S3). This is basically consistent with the AOD prediction accuracy in the Amazon by global models using the same fire emission inventory (Reddington et al., 2019; Pan et al., 2020). The slightly low bias in the AOD value could be related to an underestimated BB emission intensity due to errors in the detection of fires by satellite (Rosario et al., 2013) and/or an underestimation of the transatlantic transport from Africa (Holanda et al., 2020). Besides, the lack of SOA production in the model may also account for the bias in the AOD simulation (Bond and Bergstrom, 2006).'

**Table S3.** Comparison of AOD and SSA at 550 nm obtained from model simulation in domain3 and observation.

| | Observation | Model[a] |
|---|---|---|
| | AOD | |
| Manaus_EMBRAPA (AERONET) | 0.24±0.10 (average of Sep 2014) | 0.21±0.05 (R[b]=0.54) |
| | SSA | |
| TT34[c] (Rizzo et al., 2013) | 0.87±0.06 (average of Jul–Dec 2008–2010) | 0.89±0.01 |
| ATTO[d] (Saturno et al., 2018b) | 0.88 (average of Aug–Nov 2012–2017) | 0.90±0.01 |

a) Model results with EMIS1, averaged for September 2014.
b) R represents the correlation coefficient between the observation and model simulation.
c) The SSA values at this site are for 637 nm. Calculation of SSA at 550 nm is not conducted due to incomplete information on Angstrom exponent in Rizzo et al. (2013).
d) The SSA observation for the ATTO site is obtained from Saturno et al. (2018b) by extrapolating the original value at 637 nm to that at 550 nm using the Angstrom exponents in Saturno et al. (2018b).

*7. The authors may consider to move Section 3 to supplement to make the manuscript less length and more focused.*

**Response:** Very good suggestion. We have moved Section 3 to SI and added a short summary of the evaluation before the results section (Line 208).

'The WRF-Chem simulation with the EMIS1 scenario was evaluated for the meteorological conditions and the aerosol field using ground-based, radiosonde, and satellite remote sensing measurements (see Supplement Text S1–S3). The results show that the model simulation at 3 km resolution reasonably reproduces the metrological field in terms of surface conditions, vertical atmospheric structure, and regional precipitation. The total cloud fraction and liquid cloud amount are well captured by the model while the simulated ice water amount shows lower magnitude than the observations. The model generates close agreement of the predicted aerosol properties with the observations, including the aerosol optical properties

(AOD and SSA) and the CCN concentrations at different supersaturation conditions. Details of the model evaluation are provided in the Supplement. The satisfactory performance of the model enables it to provide reliable assessments of the BB aerosol effects on the regional climate through aerosol-radiation-cloud interactions.'

*8. Technical comments: Line 27: 'which enables them' -> 'which enable them'*

**Response:** Accepted.

*9. Technical comments: Line 151: Need a reference*

**Response:** Accepted. The reference has been added at Line 158 of the revised manuscript:

'Anthropogenic emissions were from the EDGAR-HTAPv2, a global gridded air pollution emission dataset with a resolution of $0.1° \times 0.1°$ (http://edgar.jrc.ec.europa.eu/htap_v2; Janssens-Maenhout et al., 2015).'

Janssens-Maenhout, G., Crippa, M., Guizzardi, D., Dentener, F., Muntean, M., Pouliot, G., Keating, T., Zhang, Q., Kurokawa, J., Wankmüller, R., Denier van der Gon, H., Kuenen, J. J. P., Klimont, Z., Frost, G., Darras, S., Koffi, B., and Li, M.: HTAP_v2.2: a mosaic of regional and global emission grid maps for 2008 and 2010 to study hemispheric transport of air pollution, Atmos. Chem. Phys., 15, 11411–11432, https://doi.org/10.5194/acp-15-11411-2015, 2015.

**Reference**

Becker, E. J., H. van den Dool, and M. Peña, 2013: Short-Term Climate Extremes: Prediction Skill and Predictability. J. Climate, 26, 512–531, https://doi.org/10.1175/JCLI-D-12-00177.1.

Response to Comments of Referee#2

Dear Reviewer:

We would like to thank you for the valuable and constructive comments/suggestions which helped us to improve our manuscript. We have carefully revised the manuscript accordingly. Please find our point-to-point responses below (line numbers and figure numbers refer to the new version of manuscript; reviewer comments and suggestions are in italics, responses are in plain font; revised sections in the manuscript text in response to the comments are marked with red color).

*1. To assess ARI, why not contrasting the experiment PC3_EMISX and PCNR3_EMISX? The current way to obtain ARI has an underlying assumption that that the total aerosol effects are a linear combination of ACI and ARI, which may not be the case because of the complexity of the nonlinear microphysics-dynamics-thermodynamics interactions of the system. Such an uncertainty should be discussed in the paper.*

**Response:** We thank the reviewer for this insightful comment. This paper focuses on assessing the ARI of biomass burning aerosols (BBA), but the PC3_EMISX and PCNR3_EMISX include aerosols from biomass burning origin and other sources such as anthropogenic. Contrasting the experiment PC3_EMISX and PCNR3_EMISX results in the ARI of all aerosols (biomass burning plus other sources). In the studied domain, the contribution of non-BBA to the bulk aerosol optical property, although not dominant, is noticeable, e.g. the black carbon emission rate in the EMIS1 scenario is 1.8 mg m$^{-2}$ s$^{-1}$ for biomass burning emissions and 0.4 mg m$^{-2}$ s$^{-1}$ for anthropogenic emissions, and the non-BBA proportion is even accentuated in the EMIS0.5 case. The method used in this study to assess the ARI of BBA refers to the same method used in Archer-Nicholls et al. (2016) for separating BBA's indirect effect, i.e. ACI in this study, and radiative effect (direct+semi-direct), i.e. ARI in this study. This method assumed a closure relationship between the total aerosol effect and individual effects (ARI and ACI) and calculated the ARI as all the BBA-induced perturbations except those induced by the ACI pathway. This assumption was also found conventionally applied in assessing the radiative forcing of specific aerosols by indirect, direct and semi-direct effects separately and jointly (Ghan et al., 2012). By this method, the ARI of BBA can be obtained without the influence from non-BBA since the ARI from non-BBA (CC3-CCNR3) was deducted from the ARI of all aerosols (PC3_EMISX- PCNR3_EMISX).

On the other hand, the authors acknowledge the reviewer's concern that the nonlinear nature of the cloud system may make ARI assessed in the present way different from the results by contrasting the simulations with and without BBA radiative feedback. We calculated the difference between these two definitions of ARI, based on the EMIS6 scenario, since the non-BBA proportion could be neglected at high biomass burning emission intensity (the

black carbon emission rate in the EMIS6 scenario is 10.8 mg m$^{-2}$ s$^{-1}$ for biomass burning emissions and 0.4 mg m$^{-2}$ s$^{-1}$ for anthropogenic emissions). The results (Table S1) show that the difference between these two definitions of ARI is small (within the range of standard error) and does not influence the conclusions about the relative importance of ACI and ARI in this study.

The discussion about this uncertainty has been added (Page 7, Line 199):

'Due to the nonlinear nature of the cloud system, which involves complicated microphysics-dynamics-thermodynamics feedbacks (Stevens and Feingold, 2009), the ARI effect calculated as the residual component of the aerosol total effect aside from the ACI part may be different from directly contrasting the simulations with and without the radiative effect from BB aerosols. To assess this uncertainty, we compared the ARI effect on clouds obtained here with its counterpart, i.e., the difference between PC3_EMIX and PCNR3_EMISX, which directly computes the effect associated with aerosol-radiation interactions from all aerosols, based on the EMIS6 scenario to minimize the influence of aerosols not from BB (Table S1). It shows that the uncertainty in the ARI quantification associated with the cloud nonlinear microphysics-dynamics-thermodynamics feedbacks is very small and would not have a significant influence on the ARI assessment in this study.'

**Table S1.** Monthly mean perturbations caused by the ARI of BB aerosols for the EMIS6 emission scenario.

|  | ARI in this study | PC3_EMISX - PCNR3_EMISX | difference |
| --- | --- | --- | --- |
| LWP (g m$^{-2}$) | –3.8 | –3.9 | –0.1 (3%) |
| IWP (g m$^{-2}$) | 0.26 | 0.24 | –0.02 (8%) |

*2. It is unclear how the model treats the BC aging process. According to the present model description in Section 2.1, it seems the fresh BC are immediately mixed with other types of aerosols after emission. Such a simplified treatment could result in overestimation of the BC absorption and associated radiative forcing [Wang et al., 2018; Peng et al., 2016].*

**Response:** Thanks for pointing out this issue. For calculating the aerosol optical properties, the model uses the Maxwell-Garnett mixing rule, which treats the BC as small particles distributed randomly within a mixture of the other chemical components. The BC aging process has not yet been implemented in the WRF-Chem available to the community. We have added a clarification about the treatment of BC aging in the model:

'Note that the process of BC aging (Peng et al., 2016; Wang et al., 2018) has not been implemented in the model. In the future, it would be desirable to implement BC aging (Peng

et al., 2016; Wang et al., 2018) in order to more accurately simulate the mixing state of BC-containing aerosols.'

The immediate mixing of BC with other aerosols after emission did not cause obvious overestimation of BC absorption in the studied domain, as shown from the comparable single scattering albedo (SSA) between the model output and the observation (Table. S3). This could have benefited from the improved mixing rule used here, because the Maxwell-Garnett mixing rule was proven to overcome the unrealistic absorption enhancement of BC by the direct internal mixing to some extent and was found to provide reasonable BC absorption (Bond and Bergstrom, 2006). Besides, the fact that the studied domain is away from the intensive biomass burning source and is impacted by the fire plumes transported there hours after being emitted could also lower the influence of BC aging on the studied domain. The evaluation of simulated SSA and AOD (Table S3) shows that the model can generally capture the aerosol optical features in this region, and therefore is reliable for estimating the aerosol radiative effect.

*3. According to Fig. 6, the month-long simulations include a couple of deep convective systems with heavy precipitation (Sept. 9, 17-18). For the precipitation response analyses in Fig. 15, can the authors take a further step to assess the deep convective systems and the rest separately? Maybe a threshold of 3 mm/3hr can be applied to categorize those cases.*

**Response:** Thanks for the insightful suggestion. Accordingly, we have separated the precipitation responses for deep convective systems and the rest as the reviewer suggested and added corresponding figures and discussion to the revised manuscript.

'To examine the precipitation responses at different precipitation intensities (Fig. 10), a threshold of daily maximum 3-hour accumulated precipitation exceeding 3 mm, the upper boundary of the domain averaged amount (Fig. S6), is used to distinguish the intensive precipitation grids from the light precipitation ones. High convective strength indicated by larger CAPE (Fig. 10) corresponds to intensive precipitation, whereas relatively weaker convection is associated with the light precipitation regime. Intensive precipitation shows a significant nonlinear ARI response, whereas light precipitation tends to be reduced monotonically by the ARI. The precipitation reduction by ACI at low aerosol concentration is less prominent in heavy than in light rainfall, due possibly to the dynamic feedbacks in deep convection (Rosenfeld et al., 2008). By contrast, a stronger ACI effect at larger aerosol amounts is shown in heavy precipitation as a result of the larger potential for CCN activation in strong convection (Reutter et al., 2009). The dependence of precipitation change on aerosol concentration is greater for the intensive precipitation than the light precipitation

regime, given that the precipitation responses at the EMIS1 and EMIS6 scenarios are –1% and –27 % respectively for the intensive regime and –5% and –17% respectively for the light precipitation. This is consistent with the rainfall sensitivity to increasing aerosol concentration for strong and weak convection in Chang et al. (2015). The dominance role of ARI over ACI at high aerosol loadings is found at both regimes.'

**Figure 10.** Changes in domain-averaged precipitation rate with increasing BB emission intensity (indicated by domain-averaged AOD in each emission scenario) at intensive precipitation regime (a) and light precipitation regime (b). The vertical dotted line in each plot indicates the EMIS1 scenario. Error bars denote the standard error.

*4. For the IWP evaluation, how are the satellite data are averaged spatially? It seems the satellite observations shown in Fig. 5 are averaged over cloud points only. I doubt ice heterogeneous nucleation scheme can explain such a huge discrepancy. Even if the ice production scheme is not a function of INP concentration in this microphysics, it should still exist (most of time as function of temperature).*

**Response:** Thanks for drawing attention to this point. The satellite datasets do have missing data points over the domain due to a combination of both unrecognized cloud ice signals and satellite technical problems such as the orbital gap (Remer et al., 2005). We calculated the countable proportion (the ratio of days when the ice water path data is not missing to the whole number of days in the study period) of each grid cell in the domain throughout the whole month (Figure R1). The countable proportion is approximately 0.5 with 6 out of 30 days having full data coverage. This can basically represent the magnitude of the IWP for the studied period and region. However, as the reviewer correctly points out, the elimination of the unrecognized weak ice signals would bias the observation results towards higher values and thus contribute to the discrepancy between the model and observation.

The ice production terms of the microphysical scheme used in the study include 1) homogeneous nucleation which occurs below -40°C and 2) depositional growth which is a function of temperature in the temperature range from -40°C to 0°C. With such an ice production parameterization, an underestimation of ice water content was found in Baro et al. (2018) by up to 80% and in this study by a similar magnitude (though the data quality contributes to some extent). A recent study by Su et al. (2018) found that introducing the ice nuclei source from dust particles into the microphysical scheme can improve the simulated ice water content by 15%. Analogous to dust particles, the biomass burning aerosols accompanied by biological material, soil dust, or ash particles was identified to efficiently improve the ice heterogeneous formation during the dry season (Seifert, P., et al., 2015). Based on these results, the missing parameterization of heterogeneous ice nucleation was listed in this study as one of the possible reasons for the IWP underestimation in the model.

We have reworded the discussion of the cloud ice comparison (Line 105 in SI). 'The uncertainties inherent in the satellite dataset, e.g., eliminating data points with unrecognized cloud ice, would bias the observation results towards higher values and thus to some extent account for the discrepancy between model and observation. Besides, uncertainties associated with the ice-phase microphysical processes, e.g., the lack of IN parameterization, may also be a potential reason for this discrepancy (Su et al., 2018).'

[Figure]

**Figure R1.** Fraction of countable data at each pixel throughout September 2014.

*5. Line 62-65, similarly, a recent study using satellite data shows nonlinear response of deep convective clouds to smoke aerosol in South America [Jiang et al., 2017].*

**Response:** Thanks for recommending the reference. This reference has been added.

10   *6. Out of personal curiosity, to what extent the FDDA can reduce the meteorological biases? If the authors have the model free run available, I like to see a comparison of those two.*

**Response:** In this study, FDDA was used in the outer domains to provide a more accurate meteorological boundary for domain 3. The simulated surface air temperature, relative humidity, and wind speed from the simulation of domain 2 with and without FDDA are
15   compared against the observations at the ATTO site (Fig. R2). A notable improvement can be found in the run with FDDA compared to the free run.

[Figure]

**Figure R2.** Scatter plot of surface air temperature (a), relative humidity (b) and wind speed (c) from observation and model simulation.

*7. Fig. 4, what are the two dash lines in addition to the 1-to-1 line?*

**Response:** The two dashed lines are 1:4 and 4:1 lines. We have clarified this in the caption.

'The dashed lines are 1:4, 1:1, 4:1 from top to bottom, respectively.'

10 *8. Line 326-327, it doesn't make much sense to compare a regional aerosol forcing to the global values.*

**Response**: Thanks, we agree. This comparison has been removed.

*9. Fig. 11 is discussed after Figs. 12 and 13. Better to reverse their order.*

**Response:** Accepted. The order has been reversed as the reviewer suggested.

*10. In Fig. 11 and 12, the larger updraft velocity and IWP by absorbing aerosols corroborate the thermodynamic invigoration hypothesis by Wang et al. [2013] which suggested larger CAPE above PBL due to the presence of absorbing aerosols in the lower troposphere.*

**Response:** Thanks for pointing out this connection. We have added this discussion at Line 329.

'This positive response of cloud ice and updraft velocity to ARI corresponds to the thermodynamic invigoration mechanism proposed in Wang et al. (2013) which suggested larger convective available potential energy (CAPE) above PBL could be induced by the absorbing aerosols in the lower troposphere.'

*11. Title is too long. Maybe remove "dependence of aerosol-cloud and aerosol-radiation interactions on aerosol loading".*

**Response:** We appreciate the reviewer's suggestion. We intended to use the subtitle to highlight the dependence mechanism studied in the paper and think it would help the readers to catch the key points effectively. We have shortened the title to 'Impact of biomass burning aerosols on radiation, clouds, and precipitation over the Amazon: relative importance of aerosol-cloud and aerosol-radiation interactions'


**Text S2**

**Evaluation of meteorological condition**

Figure S2 shows the time series of hourly surface meteorological variables observed at the ATTO site and corresponding simulated results from domain3 in September 2014. As the canopy effect is integrated in the land surface model (Lee et al.,

2016), the simulated meteorological variables are characterized by above-canopy properties. The variation patterns of the air temperature and RH are well captured by the model, with correlation coefficients of 0.86 and 0.78, respectively (Table S2). The surface air temperature is reproduced with a moderate overestimation of 0.2 °C (Table S2), which mainly occurs on 6 Sep, 8 Sep, and 17–18 Sep, whereas the RH exhibits an opposite bias. These significant biases occur corresponding to the missing prediction of rainfall on 6 and 18 Sep and underprediction of precipitation on 8 and 17 Sep by the model at this site (Fig. S3). This is expected since precipitation enhances surface evaporation and latent heat flux, leaves less net energy in the ground to heat the surface air (Zhuang et al., 2017), and therefore corresponds to a cool and moistened near-surface atmospheric state as can be found in the ATTO observation (Fig. S2). Consequently, the precipitation underestimation in the model was accompanied by higher biased simulated temperature and lower biased RH. This discrepancy between simulated and observed precipitation at the ATTO site could be associated with a bias of rainfall location and existence of unresolved subgrid-scale (<3 km), which will be further discussed later by comparing with the regional rainfall prediction. The wind speed from the simulation is generally lower than the observations with an average bias of –0.2 m s$^{-1}$ (Table S2). The underestimation of the surface wind speed by the model also existed extensively in previous WRF-Chem simulations, and was ascribed to uncertainties in surface drag parameterization (Tuccella et al., 2012; Zhang et al., 2015).

The vertical distribution of the meteorological variables at the Manaus site over the 30-day simulation period is compared in Fig. S4. To keep consistency, simulation outputs of temperature and RH from domain3 were interpolated to the standard levels of the radiosonde data. The CAPE and LCL, inferred from the temperature and humidity profiles from modeling and observations, are also shown. The model reproduces the air temperature profiles well. The RH generally follows the observed results below 300 hPa, while in the upper troposphere above 300 hPa a large overestimation occurs at 12:00 UTC. Similarly, an overestimation of simulated water vapor compared with MLS retrievals in the upper troposphere was found in WRF-Chem simulations of the Amazon Basin (Wu et al., 2011). The CAPE and LCL values estimated from the model agree well with that from the observations at 00:00 UTC. Noticeable differences of CAPE and LCL between model and observation of 240 J kg$^{-1}$ and 266 m, respectively, are seen at 12:00 UTC, implying a possible earlier development of the simulated planetary boundary layer (PBL) ahead of observations.

The daily retrievals of cloud fraction, total LWP, and total IWP from the MODIS Aqua measurements are used to evaluate the simulation performance for cloud properties by WRF-Chem. The domain3 simulation results are averaged over the domain area to compare with the corresponding variables from the satellite measurements, as shown in Fig. S5. The simulated total LWP, calculated as the sum of liquid cloud and rainwater, correlates well with observations with a moderate underestimation. The total IWP from the model, as the sum of cloud ice, snow, and graupel, basically shows a positive correlation with the observations. However, a large underestimation of the total IWP from the model exists compared to the remote-sensed data. The model performs relatively well for the extreme low and high IWP regimes, with values being approximately 25% of the observations. The simulation of the total IWP by the WRF model has been found to produce a seasonally averaged underestimation by up to 80% compared with satellite measurements (Baro et al., 2018). The uncertainties inherent in the satellite dataset, e.g., eliminating data points with unrecognized cloud ice, would bias the observation results towards higher

Deleted: Hence, a bias of rainfall location and existence of unresolved subgrid-scale (<3 km) convective precipitation may account for the discrepancy between simulated and observed surface temperature and RH.

values and thus to some extent account for the discrepancy between model and observation. Besides, uncertainties associated with the ice-phase microphysical processes, e.g., the lack of IN parameterization, may also be a potential reason for this discrepancy (Su et al., 2018). Generally, the total cloud fraction from the model shows a linear correlation with the observations, falling between 25%–75% of the observed values.

Figure S6 shows the time series of domain-averaged 3-hour accumulated precipitation from the domain3 simulation and corresponding TRMM measurements during September 2014. The model well captures the occurrence of rainfall measured remotely by satellite. The regional rainfall events on 6, 8, 17 and 18 Sep are well predicted by the model with a slight underestimation, which reflects a better model performance compared with the evident model underestimation of precipitation at the ATTO site on these four days. Moreover, the modelled synoptic patterns corresponding to the precipitation episodes are consistent with the NCEP reanalysis data (not shown). The well-reproduced regional synoptic and precipitation conditions in the model serve to corroborate that the precipitation underestimation at the ATTO site is likely induced by a local bias of rainfall location and neglecting precipitation of sub-grid convection by the model. Generally, the simulated precipitation is comparable with TRMM observations in terms of time variation and intensity, which illustrates the model's ability to represent the convective activity during the study period.

**Test S3**

**Evaluation of aerosol field**

Table S3 shows the comparison of the modelled AOD against the AERONET observation at Manaus_EMBRAPA, a forest reservation site representative of the central Amazon environment (Artaxo et al., 2013). The model simulation generally captures the absolute value and the temporal variation of the observed AOD, with the mean bias and correlation coefficient being -0.03 and 0.54, respectively (Table S3). This is basically consistent with the AOD prediction accuracy in the Amazon by global models using the same fire emission inventory (Reddington et al., 2019; Pan et al., 2020). The slightly low bias in the AOD value could be related to an underestimated BB emission intensity due to errors in the detection of fires by satellite (Rosario et al., 2013) and/or an underestimation of the transatlantic transport from Africa (Holanda et al., 2020). Besides, the lack of SOA production in the model may also account for the bias in the AOD simulation (Bond and Bergstrom, 2006).

The simulated single scattering albedo (SSA) is compared with observations from previous studies, as shown in Table S3. Compared with the in-situ measured SSA of 0.87±0.06 at 637 nm at the TT34 tower (Rizzo et al., 2013) in the central Amazon, a slightly higher value of 0.89±0.01 is obtained by the model simulation. Similarly, the modeled monthly mean SSA of 0.90 for the location of the ATTO site is relatively higher than an extrapolated value of 0.88 at 550 nm from multi-year observations for the dry season at the ATTO site (Saturno et al., 2018b). Given the substantial influence of BB particles on the aerosol SSA (Saturno et al., 2018b), the difference between model results and observation may be associated with the mismatched average time periods for the comparison. Generally, the simulated SSA does not deviate greatly from the observed

value, which reflects a reasonable representation of the aerosol optical characteristics in the model. In Fig. S7 we compare the time variation of black carbon mass concentration measured at the ATTO site during the simulation period with the simulation outputs. The model results are in fair agreement with the observed BC concentrations, indicating a reasonable estimate of the influence from BB on this region.

The aerosol vertical distribution is evaluated using the CALIPSO-measured monthly mean clear-sky aerosol extinction profile averaged over the domain3 (Fig. S8). The simulation data was processed to align with the observation by using outputs corresponding to the passing time of the satellite, excluding cloudy grids with a cloud criterion of 1 g kg$^{-1}$ and interpolating the extinction coefficient at 550 nm to 532 nm. The model reproduced the observed high aerosol extinction coefficient below 3 km and accurately captured the location of the two peaks at the surface and near 2 km respectively. Compared with the observation, the model overestimates the aerosol extinction above 3 km, which was also found in Wu et al. (2011). This discrepancy may be associated with an overestimated exchange between PBL and the free atmosphere by turbulent mixing and convective transport, an underestimation of precipitation scavenging, and/or an overestimated plume rise at some fire spots. Generally, the model reasonably simulated the aerosol vertical distribution, illustrating an acceptable performance of the plume rise parameterization. The ability of the model to reproduce the aerosol vertical pattern provides reliable aerosol input for investigating the aerosol-radiation-cloud interaction, given the important role of the vertical distribution of light-absorbing aerosols in affecting the aerosol radiative effect (Johnson et al., 2004).

A comparison of CCN concentrations at different supersaturations between in-situ observation and the WRF-Chem simulation for the ATTO site is presented in Fig. S9. The calculation of CCN number concentration at observed supersaturation level from model outputs followed the method in Su et al. (2010). The model results show an overall agreement in magnitude with observations for the supersaturation range of 0.2-0.5%, which represents the typical atmospheric conditions during the dry season in the Amazon (Archer-Nicholls et al., 2016). The variation of CCN number with supersaturation level matches the pattern obtained by observation (Pöhlker et al., 2018), indicating a reasonable sensitivity of aerosol activation ability to varying supersaturation situations.

**Table S1.** Monthly mean perturbations caused by the ARI effect of BB aerosols for the EMIS6 emission scenario.

| | ARI in this study | PC3_EMISX - PCNR3_EMISX | difference |
|---|---|---|---|
| LWP (g m$^{-2}$) | −3.8 | −3.9 | −0.1 (3%) |
| IWP (g m$^{-2}$) | 0.26 | 0.24 | −0.02 (8%) |

**Table S2.** Statistical indexes of the comparisons between modeled and observed surface air temperature (T), relative humidity (RH), and wind speed (WS) at the ATTO site over September 2014.

|  | MB | RMSE | r |
|---|---|---|---|
| T (°C) | 0.2 | 1.5 | 0.86 |
| RH (%) | −2.3 | 9.2 | 0.78 |
| WS (m s$^{-1}$) | −0.2 | 1.9 | 0.52 |

MB: the mean bias;
RMSE: the root mean square error;
r: the correlation coefficient.

**Table S3.** Comparison of AOD and SSA at 550 nm obtained from model simulation in domain3 and observation.

| | Observation | Model[a] |
|---|---|---|
| | AOD | |
| Manaus_EMBRAPA (AERONET) | 0.24±0.10 (average of Sep 2014) | 0.21±0.05 (R[b]=0.54) |
| | SSA | |
| TT34[c] (Rizzo et al., 2013) | 0.87±0.06 (average of Jul–Dec 2008–2010) | 0.89±0.01 |
| ATTO[d] (Saturno et al., 2018b) | 0.88 (average of Aug–Nov 2012–2017) | 0.90±0.01 |

a) Model results with EMIS1, averaged for September 2014.
b) R represents the correlation coefficient between the observation and model simulation.
c) The SSA values at this site are for 637 nm. Calculation of SSA at 550 nm is not conducted due to incomplete information on Angstrom exponent in Rizzo et al. (2013).
d) The SSA observation for the ATTO site is obtained from Saturno et al. (2018b) by extrapolating the original value at 637 nm to that at 550 nm using the Angstrom exponents in Saturno et al. (2018b).

**Table S4.** Estimates of radiative perturbation by biomass burning aerosols over the Amazon Basin in this study and from previous studies.

| | Description | Radiative perturbation (W m$^{-2}$)* | AOD | Effect | Region of Amazon Basin | Model | Reference |
|---|---|---|---|---|---|---|---|
| Clear-sky | SW at TOA | −5.6±1.7 | 0.25±0.11 | ARI | Southern | SBDART | Sena et al. (2013) |
| | SW at TOA | −3.33±0.89 | 0.67 | total | Southern | HadGEM3-GA3 | Thornhill et al. (2018) |
| | SW at TOA | [−0.7, −3.7] | 0.2–0.6 | ARI | Central | WRF-Chem | This study |
| All-sky | SW at TOA | 1.35±1.8 | 0.67 | total | Southern | HadGEM3-GA3 | Thornhill et al. (2018) |
| | LW at TOA | −3.07±1.55 | | | | | |
| | SW at surface | −5.46±1.93 | | | | | |
| | SW at TOA | −1.75 | 0.8–1.2 | ARI | Southwest | WRF-Chem | Archer-Nicholls et al. (2016) |
| | | 2.72 | 0.4–1.0 | | | | |
| | | 1.53 | 0.4–1.0 | | | | |
| | SW+LW at TOA | −4±1 | | ARI | Southern | MetUM | Kolusu et al. (2015) |
| | SW+LW at surface | −9±1 | | | | | |
| | LW at TOA | −0.12 | | total | entire | WRF-Chem | Wu et al. (2011) |
| | SW at surface | −15.9 | | | | | |
| | SW at surface | −28.23 | 0.633 | total | Southern | GATOR-GCMOM | Ten Hoeve et al. (2012) |
| | LW at surface | 8.6 | | | | | |
| | SW at surface | −10 | 0.2–0.4 | ARI | Northwest | CCATT-BRAMS | Rosario et al. (2013) |
| | SW at TOA | [−0.3, 0.6] | 0.2–0.6 | total | Central | WRF-Chem | This study |
| | LW at TOA | [0.1, 0.9] | | | | | |
| | SW at surface | [−6.7, −31.8] | | | | | |
| | LW at surface | [0.3, 1.9] | | | | | |
| | SW at TOA | [0.4, 2.0] | 0.2–0.6 | ARI | Central | WRF-Chem | This study |
| | LW at TOA | [0.1, 1.0] | | | | | |
| | SW at surface | [−5.7, −30.5] | | | | | |
| | LW at surface | [0.4, 2.0] | | | | | |

*Radiative perturbation with standard deviation or in bracket for range obtained from simulations with emission intensity of EMIS1–EMIS6.

[Figure]

Moved (insertion) [2]

**Figure S1.** Relationship of monthly mean domain-averaged cloud droplet effective radius and cloud-base CCN concentrations for all emission scenarios derived from experiments of CCNR3 and PCNR3_EMISX. The dashed line indicates the EMIS1 scenario. Error bars represent the 25th and 75th percentiles of all domain-averaged data in each simulation.

[Figure]

**Figure S2.** Time series of surface air temperature (a), relative humidity (b), and wind speed (c) from the domain3 simulation and the observations at ATTO during September 2014.

[Figure]

**Figure S3.** Time series of precipitation from observations at the ATTO site and WRF-Chem simulations during September 2014.

[Figure]

**Figure S4.** Vertical profiles of air temperature and relative humidity at standard levels, and retrieved CAPE and LCL values from radiosonde observations and WRF-Chem domain3 simulations at 00:00 UTC and 12:00 UTC at Manaus. Error bars at each pressure level represent the standard error at that level.

[Figure]

[Figure]

[Figure]

**Figure S5.** Scatter plots of cloud fraction (a), total liquid water path (b) and total ice water path (c) from WRF-Chem domain3 simulations and MODIS satellite measurements. The dashed lines are 1:4, 1:1, 4:1 from top to bottom, respectively.

[Figure]

**Figure S6.** Time series of region averaged 3-hour accumulated precipitation (mm) over domain3 from TRMM satellite observations and WRF-Chem simulations during September 2014.

[Figure]

**Figure S7.** Time series of simulated and observed black carbon mass concentrations at the ATTO site.

[Figure]

**Figure S8.** Monthly mean clear-sky aerosol extinction coefficient at 532 nm averaged over domain3.

[Figure]

**Figure 7.** Time series of AOD at 550 nm from AERONET measurements, MODIS daily retrievals, and WRF-Chem simulations for three AERONET stations. The locations of the AERONET stations are shown in Figure 1.¶

[Figure]

**Figure S9.** Monthly averaged CCN number concentrations at different supersaturations from ATTO observations and WRF-Chem simulations. Error bars represent the standard deviation.

[Figure]

[Figure]

**Figure S10.** Diurnal variation of changes in clear-sky shortwave radiation at TOA (a) and at the surface (b) due to ARI in the EMIS1 emission scenario. Error bars denote the standard error.

[Figure]

**Figure S11.** Diurnal variation of changes in all-sky longwave radiation at TOA (a) and at the surface (b) in the EMIS1 emission scenario. Error bars denote the standard error.

[Figure]

**Figure S12.** Diurnal variation of the vertical distribution of the domain-averaged difference in precipitating hydrometer (QRAIN+QSNOW+QGRAUP) concentrations caused by BB aerosols' ACI (a), ARI (b), and total effect (c) in the EMIS6 emission scenario.

[Figure]

**Moved up [2]:**
**Figure S5.** Relationship of monthly mean domain-averaged cloud droplet effective radius and cloud-base CCN concentrations for all emission scenarios derived from experiments of CCNR3 and PCNR3_EMISX. The dashed line indicates the EMIS1 scenario. Error bars represent the 25th and 75th percentiles of all domain-averaged data in each simulation.¶

[Figure]

**Figure S13.** Profiles of ARI-induced changes in snow, graupel, and super-cooled cloud water mixing ratios for emission scenarios EMIS1 (a) and EMIS6 (b).

[Figure]

**Figure S14.** ARI-induced changes in column-integrated graupel and super-cooled cloud water content with increasing BB emission intensity (indicated by the domain-averaged AOD in each emission scenario). The vertical dotted line in each plot indicates the EMIS1 scenario.